# A role for the centrosome in regulating the rate of neuronal efferocytosis by microglia in vivo

Katrin Möller*, Max Brambach[†], Ambra Villani[†], Elisa Gallo, Darren Gilmour, Francesca Peri*

Department of Molecular Life Sciences, University of Zürich, Zürich, Switzerland

*For correspondence:
katrin.moeller@uzh.ch (KM);
francesca.peri@uzh.ch (FP)

[†]These authors contributed equally to this work

Competing interest: The authors declare that no competing interests exist.

**Abstract** During brain development, many newborn neurons undergo apoptosis and are engulfed by microglia, the tissue-resident phagocytes of the brain, in a process known as efferocytosis. A hallmark of microglia is their highly branched morphology characterized by the presence of numerous dynamic extensions that these cells use for scanning the brain parenchyma and engulfing unwanted material. The mechanisms driving branch formation and apoptotic cell engulfment in microglia are unclear. By taking a live-imaging approach in zebrafish, we show that while microglia generate multiple microtubule-based branches, they only successfully engulf one apoptotic neuron at a time. Further investigation into the mechanism underlying this sequential engulfment revealed that targeted migration of the centrosome into one branch is predictive of phagosome formation and polarized vesicular trafficking. Moreover, experimentally doubling centrosomal numbers in microglia increases the rate of engulfment and even allows microglia to remove two neurons simultaneously, providing direct supporting evidence for a model where centrosomal migration is a rate-limiting step in branch-mediated efferocytosis. Conversely, light-mediated depolymerization of microtubules causes microglia to lose their typical branched morphology and switch to an alternative mode of engulfment, characterized by directed migration towards target neurons, revealing unexpected plasticity in their phagocytic ability. Finally, building on work focusing on the establishment of the immunological synapse, we identified a conserved signalling pathway underlying centrosomal movement in engulfing microglia.

## Editor's evaluation

This article is an important contribution to the microglia field and will be of interest to a broad readership in the fields of neurobiology, cell biology, and immunology. This work describes fundamental mechanisms of efferocytosis by microglia and uses impressive imaging in zebrafish, in combination with molecular manipulations, to provide compelling data of how centrosome movements synchronize with phagocytic cup formation during microglial efferocytosis of neuronal corpses in vivo.

## Introduction

The efficient removal of apoptotic cells by professional phagocytes, also known as efferocytosis, is a hallmark of many biological processes, ranging from embryonic development to tissue repair and immunity. Microglia are an example of this as these brain-resident macrophages engulf dying neurons during development when apoptotic cells are produced in large excess (*Ashwell, 1990*; *Ueno et al., 2013*). Indeed, it has been suggested that more than 50% of neurons die during brain development (*Levi-Montalcini, 1987*), raising the question of how the small microglial population copes with extensive neuronal cell death. Using the zebrafish system, we have shown that microglia become bloated

when they cannot digest and process engulfed neurons. Consequently, these bloated microglia reduce their engulfment rate and phagocytose fewer neurons (*Villani et al., 2019*), suggesting that intrinsic mechanisms must be in place to regulate uptake in these cells. Another observation supporting this view is that microglia can prematurely abort the efferocytosis of a neuron (*Mazaheri et al., 2014*). Interestingly, aborted efferocytosis does not correlate with obvious detectable features of the apoptotic target, such as apoptotic progression or distance from the phagocyte, suggesting that there are intrinsic mechanisms that limit microglia engulfment. Understanding these mechanisms and how microglia can modulate efferocytosis might help develop methods for manipulating these cells and their activities in different pathological contexts where microglia are known to become aggressive phagocytes (*Claes et al., 2021*; *Marschallinger et al., 2020*).

Microglia are known to migrate towards a brain injury via the activation of P2Y12 receptors that mediate ATP chemotaxis and promote movement towards necrotic neurons that are present at the site of damage (*Davalos et al., 2005*; *Sieger et al., 2012*). In the *Drosophila* embryo, it has been shown that such injuries can promote haemocyte migration by inducing the reorganization of the microtubule cytoskeleton with these cells extending a microtubule-based arm towards the wound (*Stramer et al., 2010*). This shows that microglia and macrophages can establish a clear polarity axis in response to injuries. In contrast, much less is known about how microglia polarize successfully towards individual apoptotic neurons that are scattered within the brain during development. Microglia have many cellular extensions, such as rapidly moving actin-dependent filopodia and microtubule-based extensions that have been shown to respond to brain injuries (*Bernier et al., 2019*; *Davalos et al., 2005*; *Nimmerjahn et al., 2005*). A crucial question in the field is how these contribute to the recognition and removal of apoptotic neurons during development.

The ability of microglia to polarize effectively towards target neurons is likely to be important for efferocytosis. Several studies have shown a key role for the centrosome in establishing cell polarity in various cellular contexts, such as the apicobasal polarity of epithelial cells, and front-rear polarity of motile cells (reviewed in *Barker et al., 2016*; *Tang and Marshall, 2012*). In cytolytic T cells, the movement of this organelle to the immunological synapse (IS) is critical for the efficient delivery of cytotoxic granules to kill infected cells (*Geiger et al., 1982*; *Martín-Cófreces et al., 2014*; *Stinchcombe et al., 2006*). Centrosomal localization is also known to regulate polarized protrusions, for instance, in developing neurons where it determines the site of axonal outgrowth (*Bellion et al., 2005*; *de Anda et al., 2010*; *de Anda et al., 2005*). In this context, the centrosome acts as the main microtubule-organizing centre (MTOC) and its placement facilitates polarized vesicular and protein trafficking (reviewed in *Meiring et al., 2020*).

In this study, we examined neuronal efferocytosis using in vivo imaging in the transparent zebrafish embryonic brain. This revealed a central role for microtubules and the centrosome in neuronal efferocytosis; microglia contact apoptotic neurons using long microtubule-dependent branches and successful engulfment is linked the targeted movement of the centrosome towards these contact sites. Doubling the number of centrosomes in microglia leads to the engulfment of more neurons, indicating that centrosomal migration into microglial branches is a rate-limiting step in neuronal efferocytosis.

## Results

### Characterization of neuronal removal by microglia using high-speed in vivo imaging

We have previously shown that at 3 days post-fertilization (dpf) the zebrafish optic tectum is characterized by high levels of neuronal apoptosis (*Casano et al., 2016*). In this brain region, there are around 30 microglia (*Figure 1A and B*) that keep an average distance of 30 µm from their nearest neighbour (*Figure 1C*). These cells have limited mobility (*Figure 1D*) and speed (0.011 ± 0.0051 µm/s; *Figure 1E*), which is comparable to that of adult microglia in the mouse (0.1–1.5 µm/min; *Zhang et al., 2016*). Each microglia is surrounded by several dying neurons (*Figure 1A and B*, *Figure 1—video 1*) that we can visualize using an established real-time reporter for neuronal apoptosis (Tg(*nbt:LexPR-LexOP:secA5-BFP*); *Mazaheri et al., 2014*). To understand how microglia remove these apoptotic neurons, we generated a microglial membrane reporter line Tg(*mpeg1:Gal4; UAS:lyn-tagRFPT*) and performed live imaging using single-plane illumination microscopy (SPIM) as this allows fast imaging

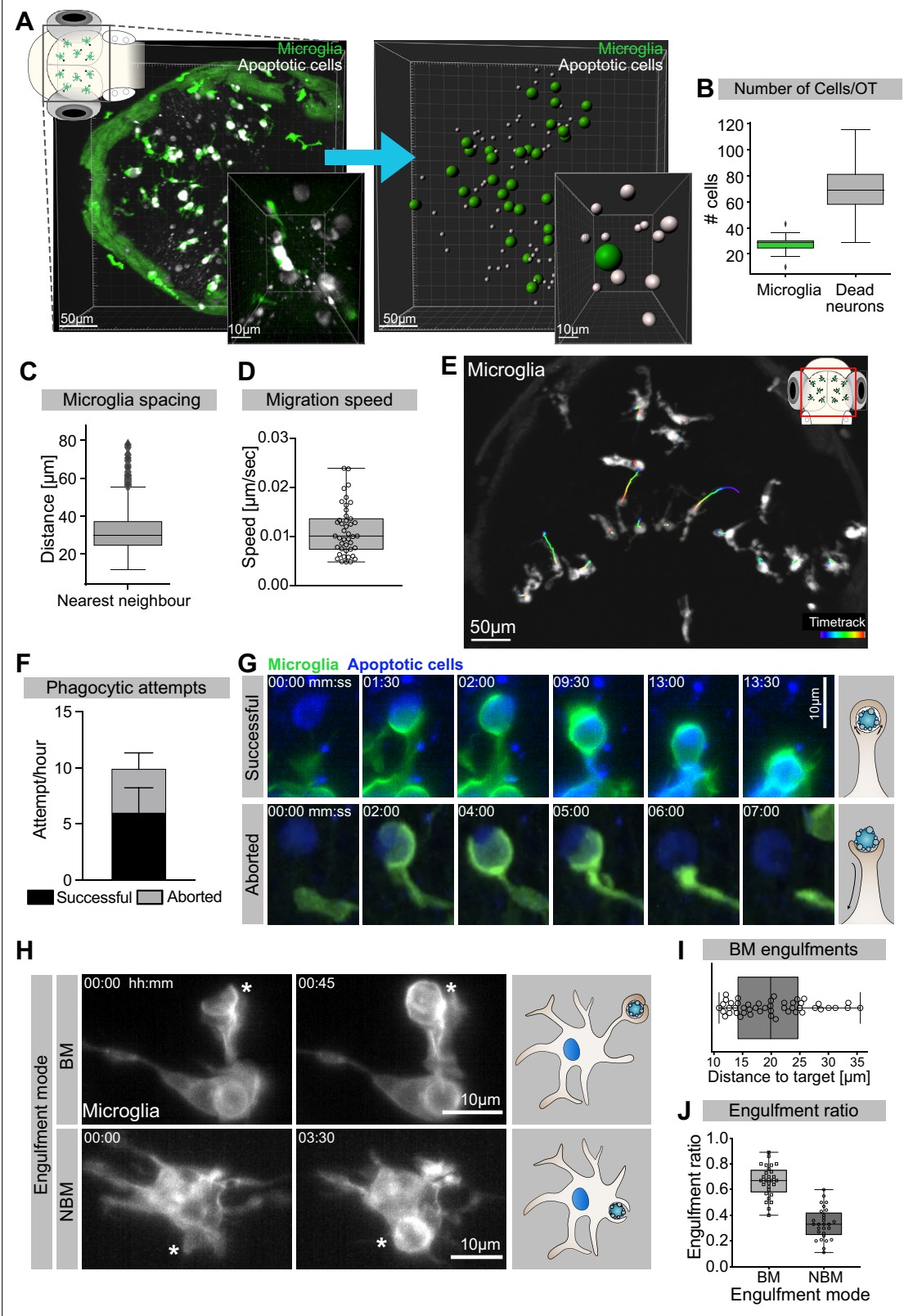

**Figure 1.** Mechanisms of neuronal efferocytosis by microglia. (**A**) Left: representative image of the OT of a 3.5-day post-fertilization (dpf) embryo with microglia (green; Tg(*mpeg1:eGFP-caax*)) and dying neurons (grey; Tg(*nbt:dLexPR-LexOP:secA5-BFP*)). Right: automated spot detection of left image to determine the spatial position of microglia and apoptotic cells. (**B**) Number of microglia and dead neurons within the OT (N = 31). (**C**) Distance from each microglia to its closest neighbour (N = 31). (**D**) Microglia migration speed, measured by tracking microglia for 2 hr (N = 2, 20–23 microglia analysed

*Figure 1 continued on next page*

*Figure 1 continued*

per zebrafish). (**E**) A representative image of a 3-dpf zebrafish brain, showing microglia (grey; Tg(*mpeg1:eGFP-caax*)) and their trajectory/track over 1 hr. (**F**) The total number of phagocytic attempts initiated by microglia per hour; stacked barplot shows the proportion of successful and aborted phagocytic attempts (N = 4, 5–9 microglia analysed per fish). (**G**) Upper panel: microglia (green; Tg(*mpeg1:eGFP-caax*)) phagocytic cup that results in the successful formation of a phagosome around a dead neuron (blue; Tg(*nbt:dLexPR-LexOP:secA5-BFP*)). Lower panel: phagocytic cup formation where the phagocytic attempt is aborted. (**H**) Microglia (grey; Tg(*mpeg1:eGFP-caax*)) phagocytosis happens at two locations; upper panel: phagosome forms at the tip of a long cellular extension. Lower panel: phagosome forms directly at the cell soma. Full time lapse is found in *Figure 1—video 2*. (**I**) Length of successful phagocytic branches during branch-mediated (BM) engulfments (N = 3, n = 7, 5–16) engulfments analysed per microglia. (**J**) Ratio between BM and non-branch-mediated (NBM) engulfments (N = 4, 5–9 microglia analysed per fish). Bars represent mean +/- SD (**F**). Boxplots depict mean and 1.5x interquartile range (**B, C**) or mean +/- min to max values (**D, F, I, J**). N refers to the number of zebrafish and n to the number of microglia examined.

The online version of this article includes the following video and source data for figure 1:

**Source data 1.** Related to *Figure 1D*.

**Source data 2.** Related to *Figure 1F and J*.

**Figure 1—video 1.** Video showing spatial position and organization of microglia and apoptotic cells.
https://elifesciences.org/articles/82094/figures#fig1video1

**Figure 1—video 2.** Time-lapse showing branch-mediated (BM) and non-branch-mediated (NBM) engulfments by microglia.
https://elifesciences.org/articles/82094/figures#fig1video2

with negligible phototoxicity (*Jemielita et al., 2013*; *Keller and Stelzer, 2008*). This approach revealed that microglia initiate on average 10 contacts per hour with surrounding apoptotic cells by forming phagocytic cups around these corpses (*Figure 1F*). Almost 60% of these cups successfully close to form a phagosome that contains the neuronal cargo (*Figure 1F and G*, upper panel, and *Video 1*), while the remaining 40% are prematurely aborted and fail to engulf (*Figure 1F and G*, lower panel, and *Video 1*). Aborted attempts have been previously described and found to be independent from the apoptotic cell and its distance from the microglia (*Mazaheri et al., 2014*). Interestingly, both successful and aborted phagocytic attempts are apparent just by having the microglial plasma membrane fluorescently labelled without the need of also imaging the apoptotic neurons; a strategy that we repeatedly used in this study to reduce imaging acquisition time (*Figure 1H*). Next, we found that microglia can engulf apoptotic neurons via two distinct mechanisms. The first mode is termed 'branch-mediated' (BM), where phagosomes form at the tip of long microglial extensions (*Figure 1H and I*, *Figure 1—video 2*) that have an average length of 20 µm (*Figure 1I*). These phagosomes are subsequently transported towards the microglial cell body (*Figure 1—video 2*). The second mode, which we term 'non-branch-mediated' (NBM), is characterized by the formation of phagosomes directly at the cell body (*Figure 1H*, *Figure 1—video 2*). This engulfment mode is employed when dead neurons are near a microglial cell. We quantified rates of BM and NBM events in unperturbed embryos and found that BM engulfment occurs more frequently (~70%; *Figure 1J*).

Thus, in the developing brain microglia are surrounded by and interact with multiple apoptotic neurons, which they remove mostly by using long cellular extensions. Interestingly, neuronal engulfment can also be unsuccessful and prematurely aborted.

## Photo-switchable destabilization of microtubules reveals phagocytic robustness and a key role for microtubules in BM phagocytosis

Having established a leading role for BM engulfments in neuronal removal, we used live reporters for microtubules (Tg (*UAS:EMTB-3xGFP*); *Revenu et al., 2014*) and F-actin (Tg(*UAS:mNG-UtrCH*); *Shkarina et al., 2021*) to determine the nature and dynamics of these extensions. We found that

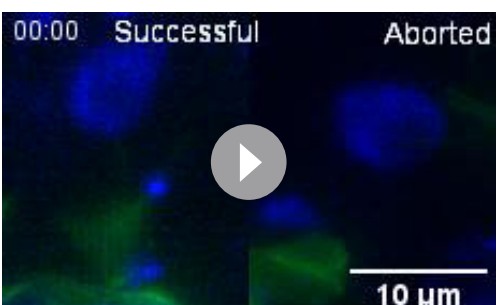

**Video 1.** Microglia phagocytic attempts can be successful or aborted. Representative microglia (green; Tg(*csf1ra:GAL4-VP16; UAS:lyn-tagRFPt*)) making successful (left) and aborted (right) phagocytic attempts. Apoptotic neurons are in blue (Tg(*nbt:dLexPR-LexOP:secA5-BFP*)). Images were acquired every 30 s using single-plane illumination microscopy (SPIM). Timescale is mm:ss.
https://elifesciences.org/articles/82094/figures#video1

microglia have different types of extensions (*Figure 2A*). They have small, actin-based filopodia that appear devoid of microtubules and are not phagocytic (*Figure 2Bii and Ci*). These move rapidly, like equivalent structures described in adult microglia (extending at 11.78 ± 0.93 µm/min and retracting at 8.11 ± 0.81 µm/min; *Figure 2D*). Microglia also have bigger branches that contain both actin and microtubules (*Figure 2Bi and Cii*). These are slower (extending at 6.16 ± 0.80 µm/min and retracting at 3.61 ± 0.82 µm/min; *Figure 2D*) and can be phagocytic (*Figure 2C and E*, *Video 2*). Microtubules extend into these branches and surround newly formed phagosomes (*Figure 2E*, *Video 2*). As previously described, upon phagocytosis there is also strong actin polymerization around the phagosome (*Figure 2F*, *Figure 2—video 1*; *Mazaheri et al., 2014*).

While many studies have revealed a key role for the actin cytoskeleton during phagocytosis (reviewed in *Mylvaganam et al., 2021*), the involvement and function of microtubules has mostly been neglected. To address this in vivo, we used photostatin (PST-1), a photo-switchable microtubule-destabilizer that allows temporally controlled perturbations during imaging (*Borowiak et al., 2015*). First, we confirmed that PST-1 can be used to depolymerize microtubules in vivo by treating zebrafish embryos at various stages and monitoring microtubule dynamics directly using an EB3 plus-end live-reporter (*EB3-mScarlet-I*; *Figure 3—figure supplement 1*; *Stepanova et al., 2003*). We imaged embryos without activating PST-1 using 561 nm illumination (referred to as PRE-PST-1) and microtubules behave normally, similar to DMSO controls (*Figure 3—figure supplement 1*). However, PST-1 activation using 405 nm exposure prevents microtubules polymerization (referred to as PST-1 ON; *Figure 3—figure supplement 1*). This illumination protocol alone does not affect microtubules in control DMSO-treated embryos, confirming PST-1 specificity (*Figure 3—figure supplement 1*). Next, we incubated embryos with labelled microglia with PST-1 for 1.5 hr during the previously described period of high apoptosis (*Casano et al., 2016*). We first imaged microglia at 561 nm (PRE-PST-1) and then induced microtubule destabilization by using 405 nm illumination (PST-1 ON; see experimental setup in *Figure 3A*). PRE-PST-1 microglia are indistinguishable from untreated cells, have a normal morphology (*Figure 3B* compared with *Figure 1H*), limited mobility (*Figure 3D* compared with *Figure 1E*), and speed (*Figure 3E* compared with *Figure 1D*). However, soon after 405 nm exposure, PST-1 ON microglia undergo a drastic morphological change characterized by loss of cellular extensions (*Figure 3B*, *Video 3*), acquisition of an amoeboid morphology (*Figure 3B and C*, *Video 3*) and increased motility (*Figure 3D and E*, *Figure 3—video 1*). Interestingly, PST-1-mediated perturbations can be reversed with 488 nm exposure (referred to as PST-1 OFF; *Figure 3F*; *Borowiak et al., 2015*). Indeed, upon 488 nm illumination, amoeboid microglia stop migrating (*Figure 3G*) and regain their typical branched morphology (*Figure 3F*).

Next, we wanted to determine how impairing microtubule dynamics affects microglial phagocytosis. Interestingly, we found that PST-1 ON microglia can still phagocytose at a rate that is comparable to that of PRE-PST-1 microglia and DMSO controls (*Figure 3H*). Interestingly, PST-1 ON microglia are unable to form branches (*Figure 3I*) and instead capture neurons by migrating directly towards targets as shown by cell tracking (*Figure 3K*) and their mean square displacement alpha (MSDα; *Figure 3J*). Treating microglia with nocodazole, a compound commonly used for depolymerizing microtubules in vivo and in vitro, revealed that these cells behave like PST-1 ON microglia (*Figure 3—figure supplement 2A–C*), confirming the role of microtubules in this behavioural switch.

Taken together, these results highlight two important roles for microtubules in neuronal engulfment by microglia. The first is that typical microglial BM engulfment depends on dynamic microtubules, and the second that when microtubule dynamics are perturbed, microglia switch to an alternative mode of engulfment –characterized by directed migration towards target neurons – revealing an unexpected level of plasticity.

## Targeted movement of the microglial centrosome into branches predicts successful neuronal engulfment

We have shown that in the presence of normal microtubule dynamics microglia favour BM engulfment (*Figure 1G*). Interestingly, we found that while during development these cells are highly branched (*Figure 2A*) and surrounded by many apoptotic neurons (*Figure 1A and B*), they only engulf one neuron at a time (*Video 4*, *Figure 4A*).

This points to the existence of intracellular mechanisms that allow branch selection in microglia. Furthermore, by examining successive phagocytic events, we found that variables such as space and

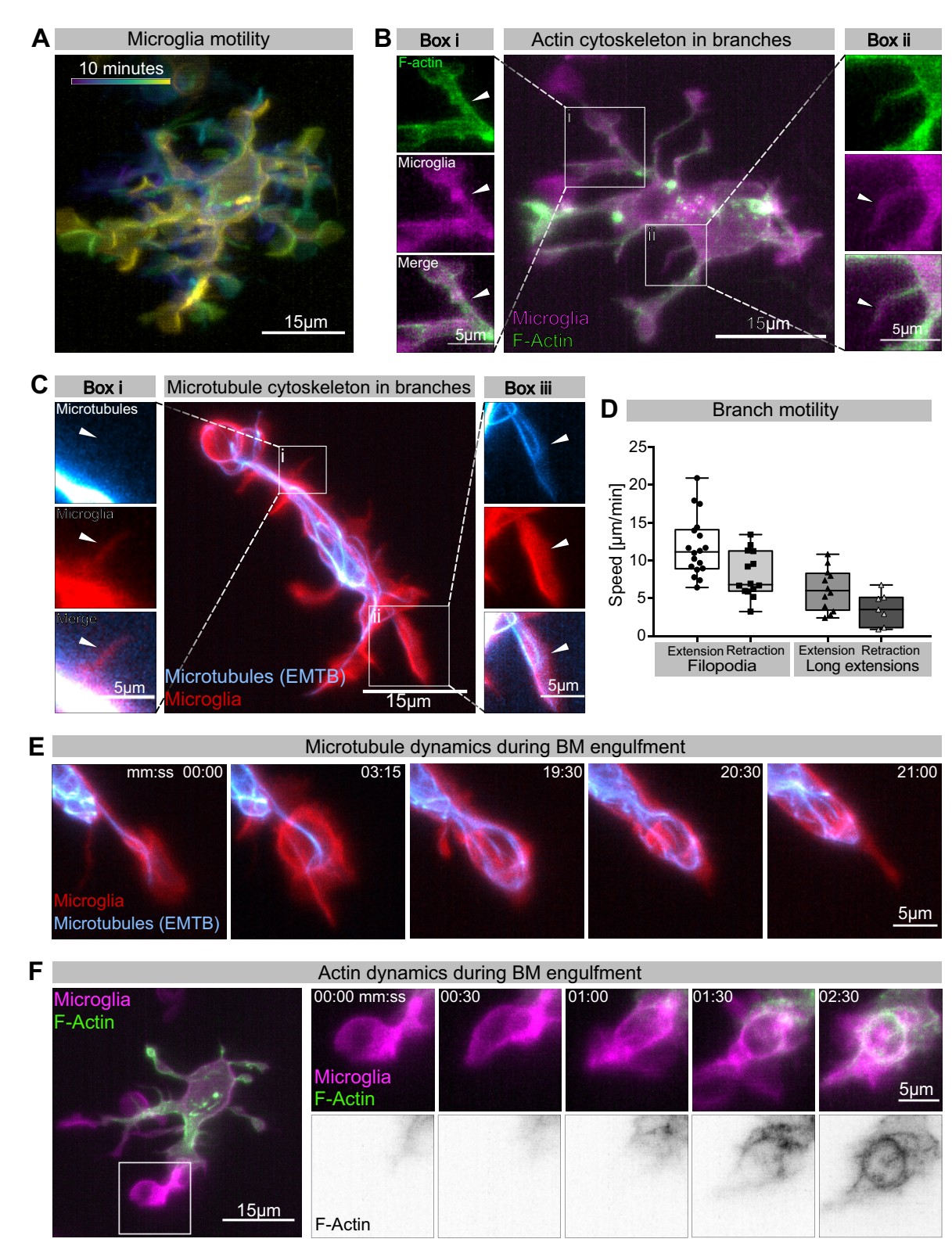

**Figure 2.** Cytoskeletal dynamics during microglia efferocytosis. (**A**) Colour-coded overlay of a 10 min time lapse of microglia Tg(*mpeg1:Gal4; UAS:lyn-tagRFPt*) showing their branch dynamics. (**B**) Microglia (magenta; Tg(*mpeg1:Gal4; UAS:lyn-tagRFPt*)) and F-actin (green; *UAS:UtrCH-mNG*). Box i shows thicker extensions, and box ii shows thin filopodia, both containing F-actin. (**C**) Microglia (red; Tg(*csf1ra:GAL4-VP16; UAS:lyn-tagRFPt*)) and microtubules (cyan; Tg(*UAS:EMTB-3xGFP*)). Box i shows a thin filopodia, and box ii shows a thicker extension; microtubules are only observed in thicker extensions.

*Figure 2 continued on next page*

*Figure 2 continued*

The intensity histogram of images in boxes i and ii from (**B**) and (**C**) has been adjusted to highlight the weak fluorescent signals of the cytoskeletal components. (**D**) The speed at which both thick extensions and thin filopodia extend and retract, measured in µm/min (n = 2, 2–10 retractions/extensions analysed per microglia, boxplot depict mean +/- min to max values). (**E**) The panel shows the same branch as in (**C**) box ii at a later time point, forming a phagosome. (**F**) Microglia (magenta; Tg(*mpeg1:Gal4; UAS:lyn-tagRFPt*)) and F-actin (green; (*UAS:UtrCH-mNG*)). The panel shows how F-actin is found in the cellular extension and engages with the phagosome soon after this has formed. Full time lapse is found in *Figure 2—video 1*. n refers to the number of microglia examined.

The online version of this article includes the following video and source data for figure 2:

**Source data 1.** Related to *Figure 2D*.

**Figure 2—video 1.** Time-lapse showing F-actin dynamics during microglia branch-mediated engulfment.

https://elifesciences.org/articles/82094/figures#fig2video1

time are correlated (*Figure 4B*); that is, the time delay between two consecutive phagocytic events is larger if apoptotic neurons are further apart from each other relative to the microglia cell. This indicates that the mechanisms that facilitate microglial polarization towards apoptotic neurons are likely to be rate-limiting and determine the speed of phagocytosis in these cells.

One component known to play critical roles in the organization and polarization of cells is the centrosome. To investigate centrosomal behaviour during phagocytosis, we generated a live reporter for the centrosome by fluorescently labelling *centrin 4*, a core component of the centrosome (Tg(*UAS:miRFP670-cetn4*); *Norden et al., 2009*). We then tracked the microglial centrosome in 3D and found that in these cells this organelle is highly dynamic, moving with an average speed of around 2 µm/min (*Figure 4C*) and undergoing rapid changes in position (*Figure 4C–E*, *Video 5*). It can be seen translocating from the cell soma into individual microglial branches, a movement that coincided with successful BM engulfments (*Figure 4D and E*, *Video 5*). In such conditions, the centrosome usually moves faster, suggesting that it accelerates towards forming phagosomes (*Figure 4C and D*). To quantify centrosomal movement over many samples, we tracked the position of the centrosome relative to the centre of the cell and successful phagocytic events (*Figure 4Fi–iii*). This analysis showed that centrosomal migration predicts successful phagocytosis as the centrosome begins moving and aligning towards the target before its engulfment (*Figure 4Fi–ii and Hi*). Following this, both the phagosome and centrosome move back towards the microglial cell soma (*Figure 4Fiii*). Interestingly, we also found that the centrosome neither aligns nor moves towards aborted phagocytic attempts (*Figure 4H*) and its position relative to these events appears to be random (*Figure 4G and H*). Thus, centrosomal migration provides a means to distinguish successful engulfment from aborted attempts.

While we occasionally saw the centrosome in the vicinity of NBM engulfments (example in *Figure 4E*), in most cases its position was random compared to BM engulfments (*Figure 4—figure supplement 1A*), indicating that targeted migration of this organelle is dispensable for phagosomes that form directly at the cell soma. The position of the centrosome was also found to be random in nocodazole-treated microglia that lack cellular branches (*Figure 3—figure supplement 2A*, *Figure 4—figure supplement 1B*), further supporting a model where centrosomal targeting is involved in successful branch selection.

Together, these observations suggest a scenario in which microglia can sense and contact apoptotic neurons using their branches; however,

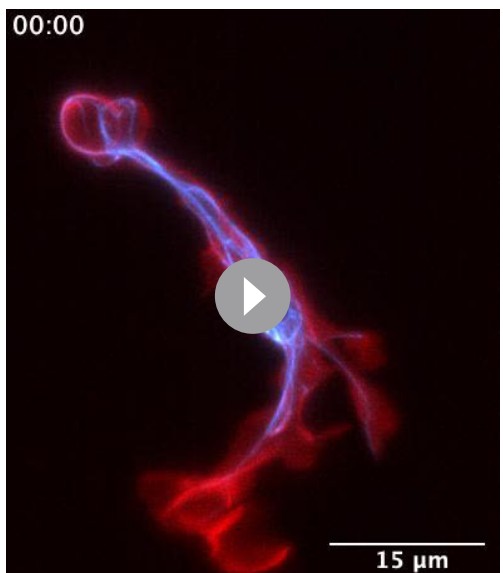

**Video 2.** Microtubule dynamics during microglia branch-mediated engulfment. Representative microglia (red; Tg(*csf1ra:GAL4-VP16; UAS:lyn-tagRFPt*)) expressing a microtubule reporter (cyan; Tg(*UAS:EMTB-3xGFP*)). Images were captured every 15 s for 32 min using single-plane illumination microscopy (SPIM). Timescale is mm:ss.
https://elifesciences.org/articles/82094/figures#video2

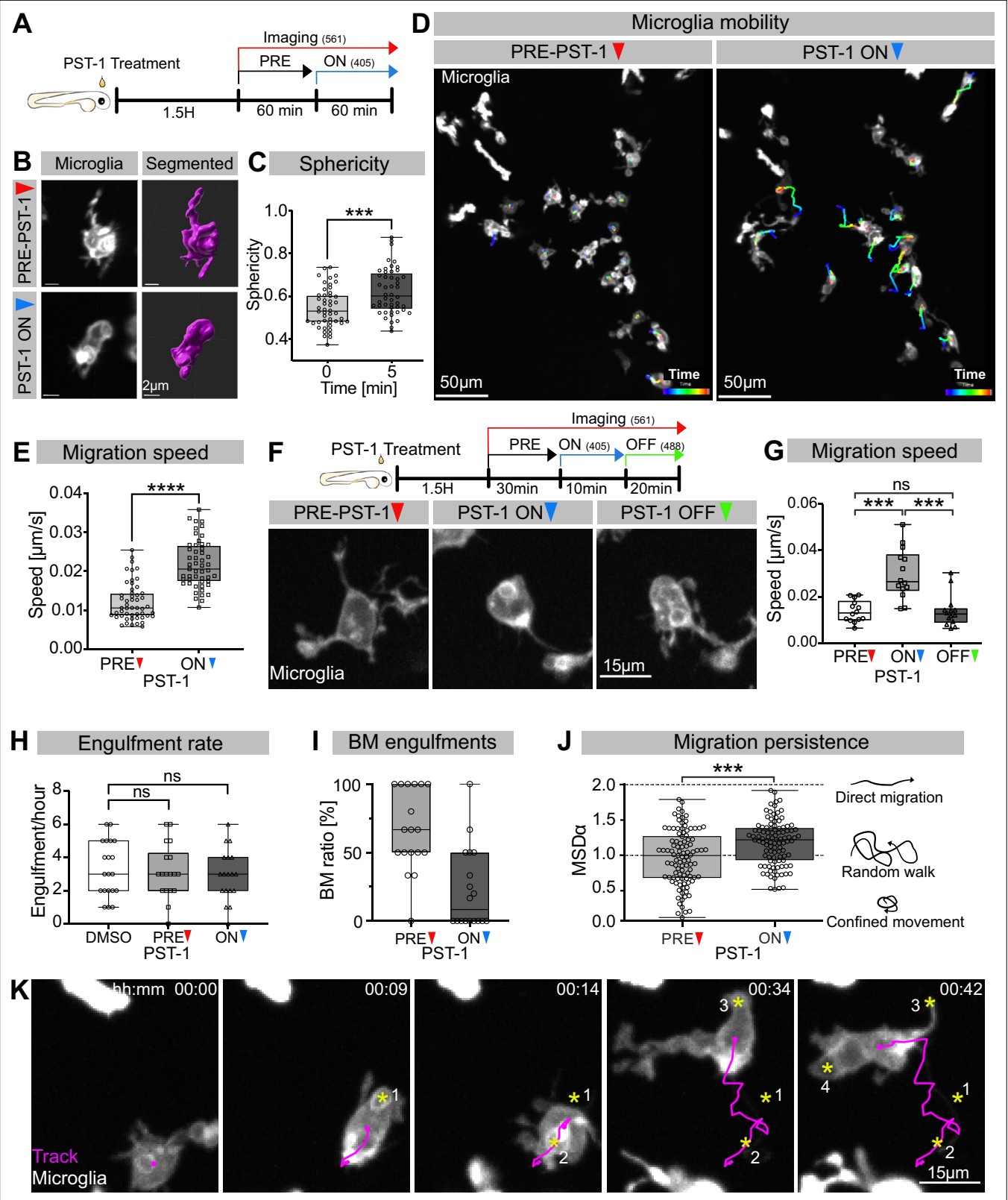

**Figure 3.** Microglia without dynamic microtubules adopt a different engulfment strategy. (**A**) Schematic showing the experimental setup: Embryos were treated for 1.5 hr with PST-1, then imaged for 1 hr with 561 nm light (PRE-PST-1) and then imaged for another hour with 561 nm and 405 nm light to activate PST-1 (PST-1 ON). (**B**) Left: microglia (grey; Tg(*mpeg1:Gal4; UAS:lyn-tagRFPt*)) before (upper) and after (lower) PST-1 activation. Right: 3D segmentation of the cells in the left panel. (**C**) Microglia sphericity before and 5 min after PST-1 activation. Segmentation in (**B**) was used to extract

*Figure 3 continued on next page*

*Figure 3 continued*

surface area and volume to calculate sphericity (N = 4, 6–15 microglia analysed per fish, ***p=0.0005). (**D**) 3D tracking of microglia for 1 hr before (left) and after (right) PST-1 activation, displayed as colour-coded tracks overlaid on top of microscopy images of the microglia. Full time lapse is found in *Figure 3—video 1*. (**E**) Microglia migration speed measured in µm/s (N = 4, 6–18 microglia analysed per fish, ****p<0.0001). (**F**) Upper: schematic of the ON-OFF experiment. Lower: microglia (grey; Tg(*mpeg1:Gal4; UAS:lyn-tagRFPt*)) shown before PST-1 activation (PRE-PST-1) and when the drug is turned ON with 405 nm light (PST-1-ON) and OFF again using 488 nm light (PST-1-OFF). A paired, non-parametric Friedman test with Dunn's correction was used to compare the three groups (***p=0.0002). (**G**) Microglia migration speed during the ON-OFF experiment, measured in µm/s (N = 1, n = 14). (**H**) Microglia engulfment rate after DMSO (N = 3, 5–8 microglia analysed per fish) or PST-1 treatment (N = 3, 6 microglia analysed per fish). (**I**) Percentage of branch-mediated (BM) engulfments before (left) and after (right) activation of PST-1 (N = 3, 6 microglia analysed per fish). (**J**) Left: microglia migration persistence measured with the mean square displacement alpha (MSDα) before and after PST-1 activation (N = 4, 6–18 microglia analysed per fish, ***p=0.0001) Right: schematic of what the MSDα values imply. (**K**) Representative time lapse of a microglia (grey) treated with activated PST-1 and tracked (magenta), showing how microglia migrate towards apoptotic targets (yellow asterisk). Boxplots depict mean +/- min to max values. Groups were compared using a two-tailed, nonparametric Mann-Whitney U-test was used (C, E, H, J). When comparing reapeated measurements from the same group, a nonparametric Friedmann test with a Dunn's correction was used (G). N refers to the number of zebrafish and n to the number of microglia examined.

The online version of this article includes the following video, source data, and figure supplement(s) for figure 3:

**Source data 1.** Related to *Figure 3C*.

**Source data 2.** Related to *Figure 3E*.

**Source data 3.** Related to *Figure 3G*.

**Source data 4.** Related to *Figure 3H and I*.

**Figure supplement 1.** Validation of PST-1 functionality in vivo.

**Figure supplement 2.** Nocodazole and PST-1-treated microglia behave similarly.

**Figure supplement 2—source data 1.** Related to *Figure 3—figure supplement 2B and C*.

**Figure 3—video 1.** Microglia without dynamic microtubules become more mobile.

successful engulfment is linked to the targeted movement of the centrosome towards these contact sites. The centrosomal position might thus explain the 'one successful branch at a time' behaviour that we observed in engulfing microglia, suggesting that movement of this organelle might be a key mechanism for limiting engulfment in these cells.

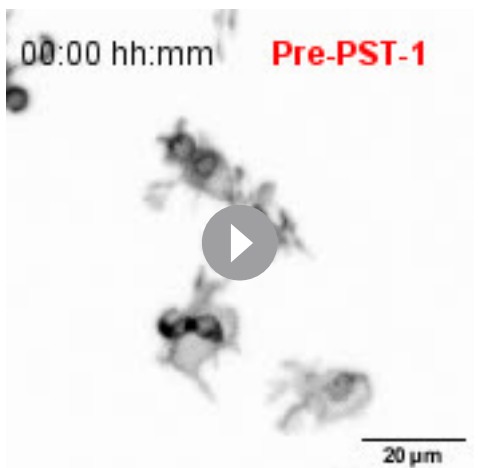

**Video 3.** Without dynamic microtubules, microglia become amoeboid and mobile. Representative time lapse of microglia (black; Tg(*mpeg1:Gal4; UAS:lyn-tagRFPt*)) treated with PST-1, before (Pre-PST-1; first hour), and after (PST-1-ON; second hour) PST-1 activation using 405 nm light. Images were captured every minute for 2 hr using spinning-disc microscopy. Timescale is hh:mm.

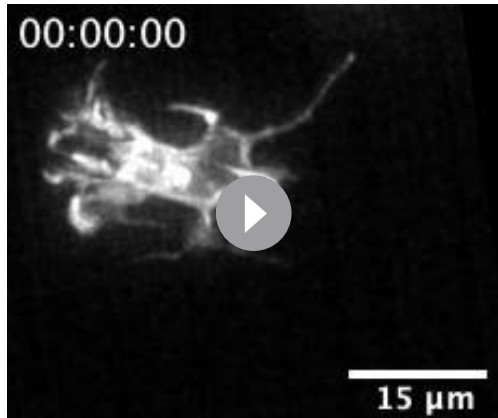

**Video 4.** Microglia engulf dead neurons sequentially. Time lapse of a representative microglia (grey; Tg(*csf1ra:GAL4-VP16; UAS:lyn-tagRFPt*)) capturing apoptotic neurons (not labelled) sequentially. Phagocytic events are marked with an asterisk. Images were captured every 30 s for 1,5 hr using single-plane illumination microscopy (SPIM). The original time lapse was deconvolved using Huygens deconvolution. Time scale is hh:mm:ss.

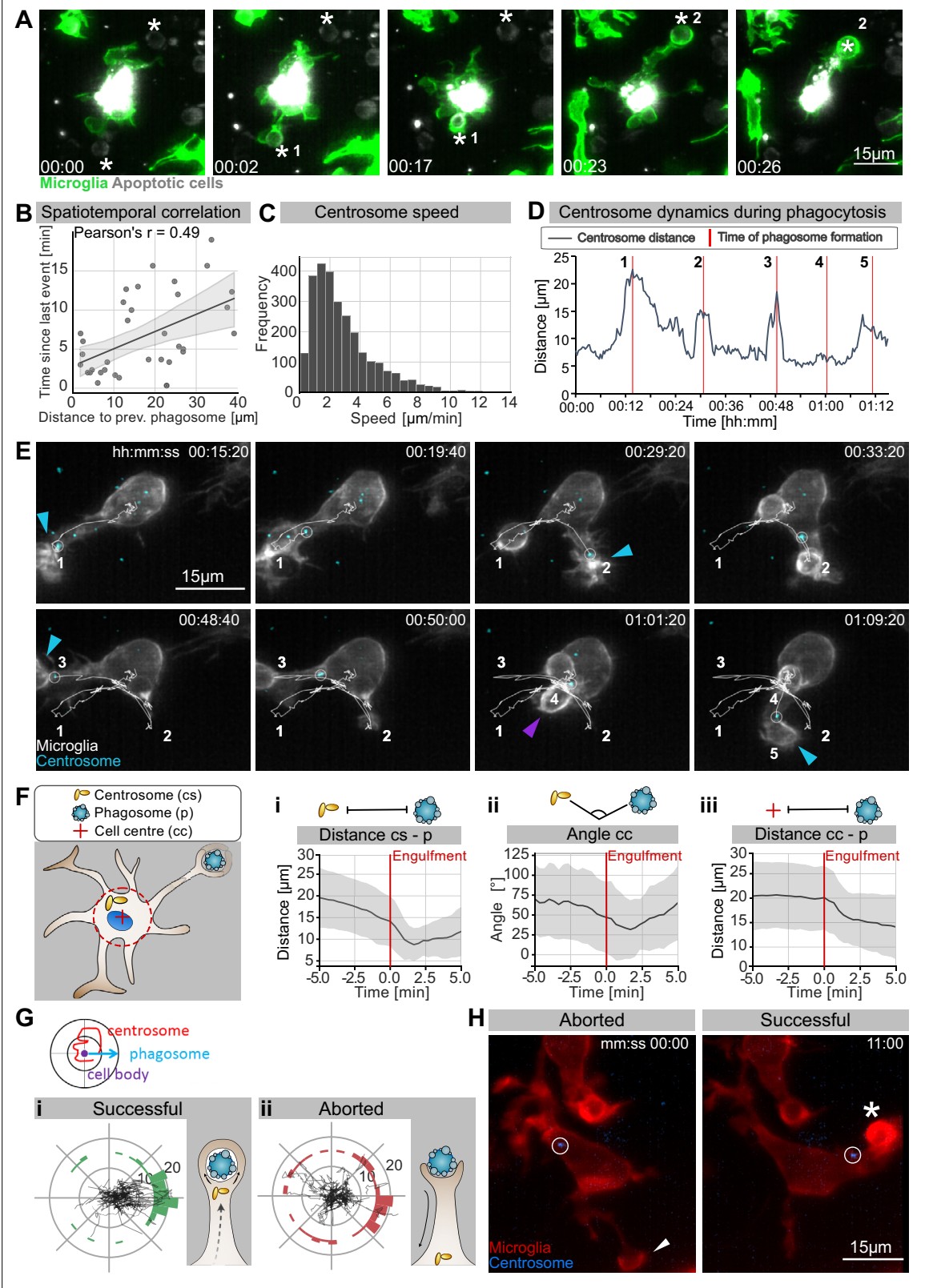

**Figure 4.** The microglia centrosome is dynamic and moves into successful phagocytic branches. (**A**) Sequential uptake of two apoptotic neurons (grey; Tg(*nbt:secA5-BPF*)) by microglia (green; Tg(*mpeg1:Gal4; UAS:lyn-tagRFPt*)). Timescale is hh:mm. (**B**) Correlation of spatiotemporal distances between phagocytic events (N = 3, n = 7, 5–16 engulfments analysed per microglia, 95% confidence interval is depicted). (**C**) Speed of centrosomal movements, relative to the cell centre (N = 3, 2–3 microglia analysed per fish). (**D**) Movement of the centrosome, relative to the cell centre of the microglia in (**E**).

*Figure 4 continued on next page*

*Figure 4 continued*

The track shows the distance of the centrosome from the cell centre, and the red lines indicate the time of phagosome formation. (**E**) Time lapse of a microglia (grey; Tg(*csf1ra:GAL4-VP16; UAS:lyn-tagRFPt*)) and the centrosome (cyan; Tg(*UAS:miRFP670-cetn4*)). The microglial centrosome is encircled and tracked for 1 hr 15 min; phagocytic events are numbered and labelled; branch-mediated (BM) engulfments are indicated with a blue arrow and non-branch-mediated (NBM) engulfments are indicated with a purple arrow. Full time lapse is found in **Video 5**. (**F**) Schematic of how the centrosome (cs), cell centre (cc), and location of newly formed phagosomes (p) were tracked across many samples. (**i**) cs distance from cb, (**ii**) angle between cs and p, and (**iii**) distance of p from cc for 5 min before and after phagosome formation (N = 7, n = 7, 5–16 engulfments analysed per microglia, the mean +/- SD is depicted). (**G**) Top: schematic of how the centrosome, cell centre, and location of a newly formed phagosomes are oriented in the polar plots. Bottom: individual tracks of the centrosome relative to the cell body and the phagosome during (**i**) successful and (**ii**) aborted phagocytic attempts (N = 3, n = 7, 5–16 successful and 2–12 aborted attempts analysed per microglia). (**H**) Microglia (grey) and the centrosome (cyan) where the centrosome is encircled during an aborted (left, arrowhead) and a successful event (right, asterisk). Original image was processed with a Gaussian Blur filter with sigma(radius) = 1. N refers to the number of zebrafish and n to the number of microglia examined.

The online version of this article includes the following source data and figure supplement(s) for figure 4:

**Source data 1.** Related to *Figure 4D*.

**Figure supplement 1.** The microglia centrosome only moves towards branch-mediated (BM) phagosomes.

## Movement of the centrosome into microglial branches correlates with targeted vesicular trafficking towards forming phagosomes

Having established a link between centrosomal movements and phagocytosis in microglia, we wanted to determine whether the centrosome acts as the main MTOC in these cells. By using the plus-end microtubule EB3 reporter in microglia Tg(*UAS:EB3-GFP*) or Tg(*UAS:EB3-mScarlet-I*), we can visualize the MTOC as a dense cluster of comets (*Figure 5A*, *Figure 5—video 1*). This EB3 cluster colocalizes with the centrosomal reporter, confirming the MTOC role for the centrosome in microglia (*Figure 5A*). Live imaging also showed that the EB3-labelled MTOC behaves like the centrosomal centin marker; it translocates into branches prior neuronal engulfment (*Figure 5B*, *Figure 5—video 2*) but not during aborted attempts (*Figure 5B*). Upon MTOC translocation, microtubules can be seen growing in the direction of phagosomes, forming a clear polarity axis and surrounding these vesicles (*Figure 5B*, *Figure 5—video 2*). This polarity axis was also observed using another microtubule reporter (Tg(*UAS:EMTB-3xGFP*); *Figure 2C and E*, *Video 2*).

It is well-known that phagosomes mature by fusing with endosomes and lysosomes (reviewed in *Levin et al., 2016*). We wondered how this is achieved in highly branched and dynamic microglia and if, for example, endosomes and lysosomes migrate into branches towards newly formed BM phagosomes. Thus, we generated reporters for early endosomes Tg(*UAS:mNeonGreen-Rab5*) and lysosomes Tg(*UAS:Lamp1-mGFP*) and established that the expression of these markers does not affect microglial phagocytosis by counting the number of phagosomes and their size in transgenic and control microglia (*Figure 6—figure supplement 1A–F*). Live imaging showed that lysosomes remained mostly within the cell soma (*Figure 6A*), they do not move into branches and fuse with phagosomes once these arrive in the cell soma (average time around 10 min; *Figure 6A and B*). By contrast, Rab5-positive endosomes form a coherent 'cloud' that moves rapidly within the microglial cell (*Figure 6C*, *Figure 6—video 1*). Interestingly, this Rab5 cloud always follows the centrosome,

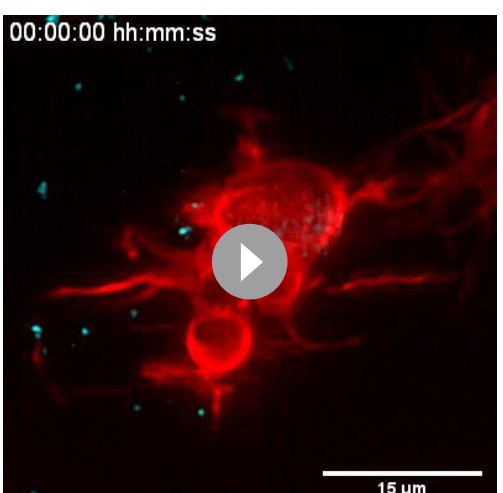

**Video 5.** The microglia centrosome is motile and moves into phagocytic branches. Representative microglia (red; Tg(*csf1ra:GAL4-VP16; UAS:lyn-tagRFPt*)) expressing the centrosome reporter (cyan; Tg(*UAS:miRFP670-cetn4*)). The microglial centrosome is marked with a circle as centrosomes from surrounding cells can also be seen in the projection. Images were captured every 20 s for 1 hr 40 min using single-plane illumination microscopy (SPIM). The original time lapse was deconvolved using Huygens deconvolution. Time scale is hh:mm:ss.

https://elifesciences.org/articles/82094/figures#video5

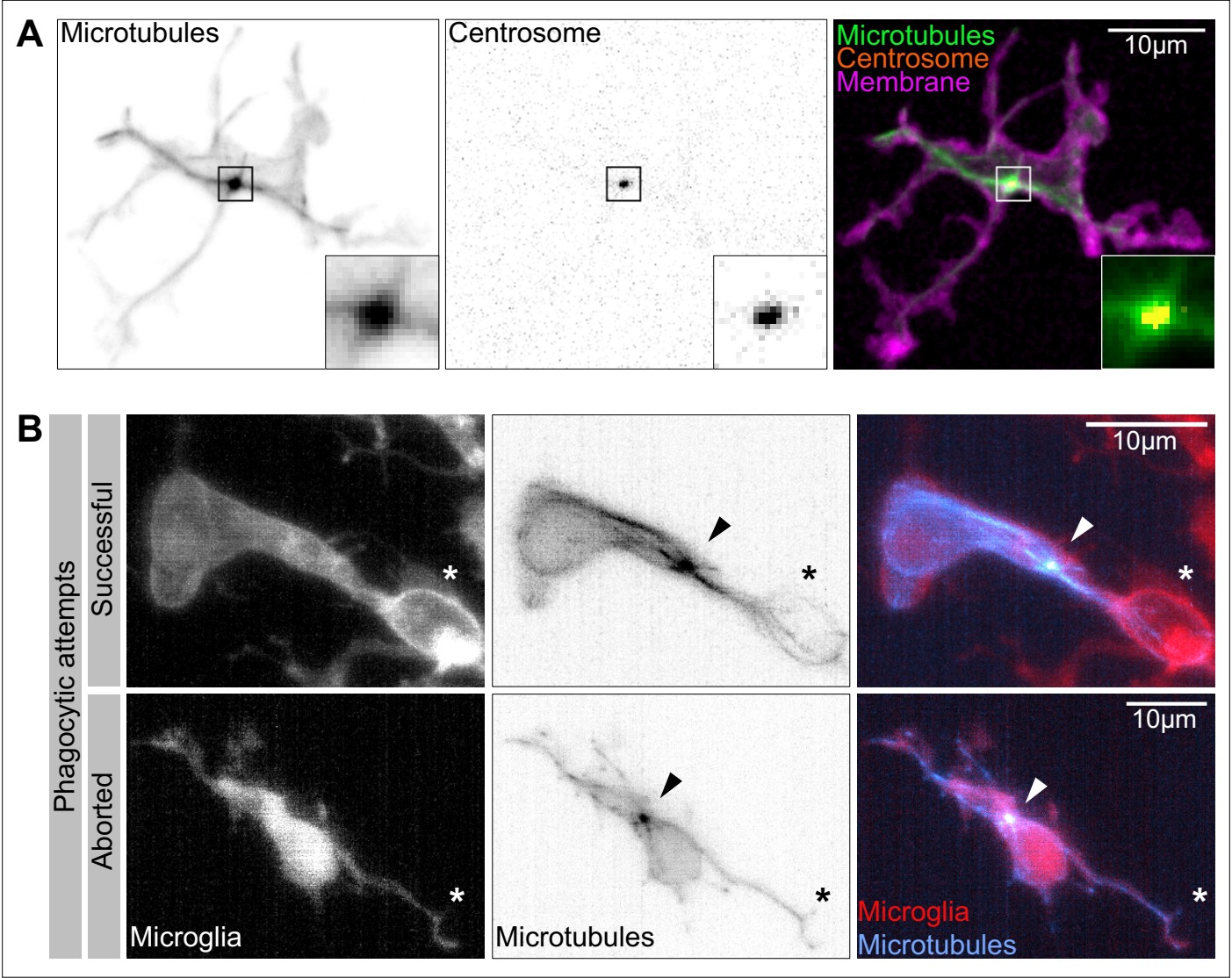

**Figure 5.** The microglia centrosome is a microtubule organization centre (MTOC) and moves into successful phagocytic branches. (**A**) Macrophage (magenta; Tg(*csf1ra:GAL4-VP16; UAS:lyn-tagRFPt*)) with the centrosome (*UAS:miRFP670-cetn4*) and microtubules (green; Tg(*UAS:EB3-GFP*)) labelled. The centrosome can be seen right in the centre of the MTOC (box). (**B**) Microglia (grey; Tg(*csf1ra:GAL4-VP16; UAS:lyn-tagRFPt*)) with microtubules (cyan; Tg(*UAS:EB3-GFP*)) labelled. The MTOC can be seen as a bright spot of growing microtubules (arrow). Top: a successful phagocytic event, full time lapse in *Figure 5—video 2*. The newly formed phagosome is marked with an asterisk. Bottom: an aborted attempt marked with an asterisk.

The online version of this article includes the following video for figure 5:

**Figure 5—video 1.** The microglia microtubule organization centre (MTOC) visualized as a dense cluster of EB3 comets.
https://elifesciences.org/articles/82094/figures#fig5video1

**Figure 5—video 2.** The microglia microtubule organization centre (MTOC) moves into successful phagocytic branches.
https://elifesciences.org/articles/82094/figures#fig5video2

travelling into phagocytic branches (*Figure 6D*, *Video 6*) and fusing with phagosomes soon after their formation (average time 5 min; *Figure 6B and C*).

These data show that centrosomal reorientation in microglia correlates with the establishment of a clear polarity axis within the cell and targeted vesicular movement towards distant forming phagosomes. This cellular compass is highly dynamic allowing distinct responses to apoptotic stimuli coming from different directions.

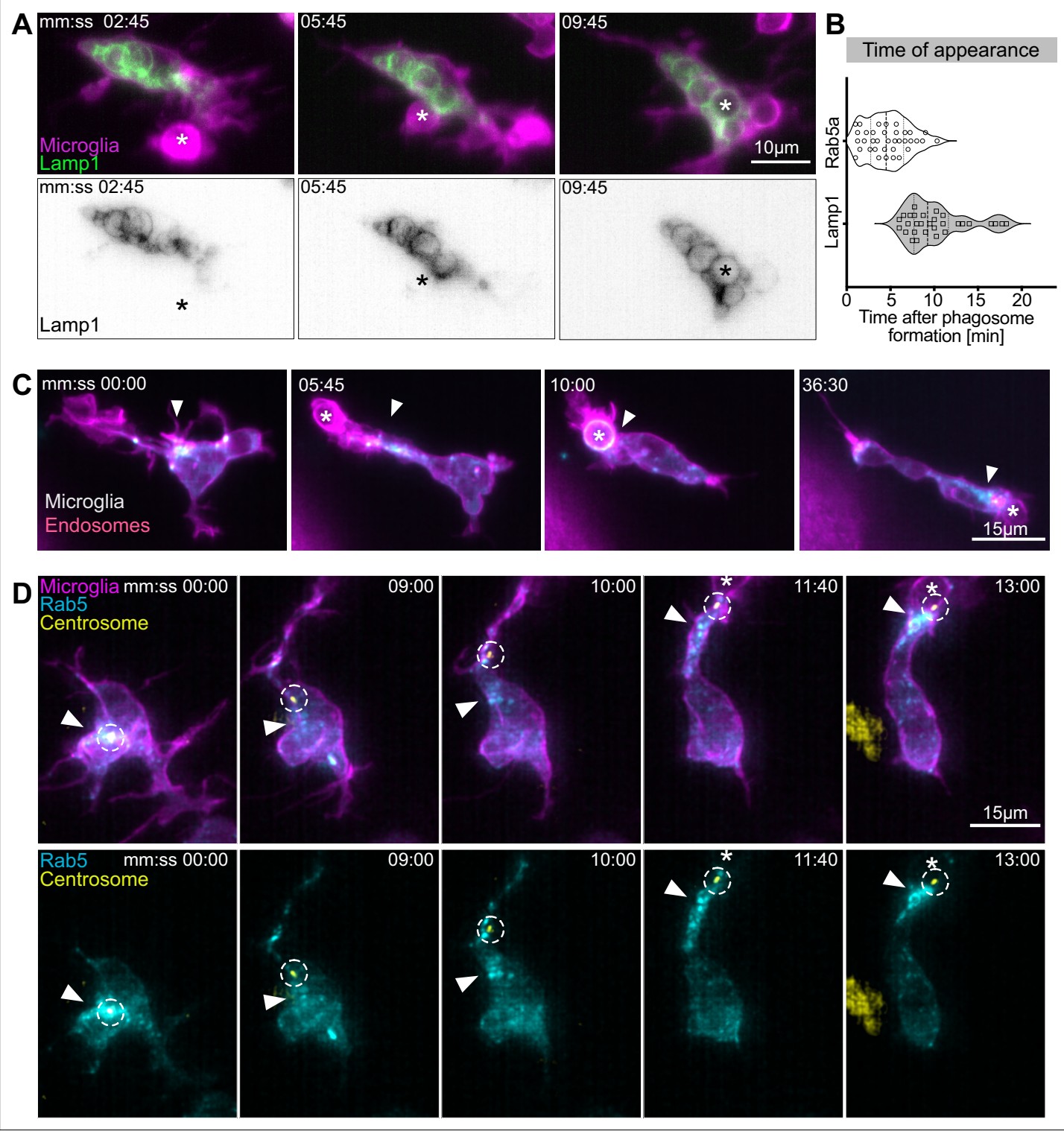

**Figure 6.** Movement of the centrosome correlates with targeted vesicular trafficking towards phagosomes. (**A**) Microglia (magenta; Tg(*csf1ra:GAL4-VP16; UAS:lyn-tagRFPt*)) and Lamp1 labelled vesicles (green; Tg(*UAS:Lamp1-mGFP*)) localized within the cell soma and not moving into branches. (**B**) Time of Rab5 (n = 3, 8–10 phagocytic events analysed per microglia) and Lamp1 (n = 3, 9–11 phagocytic events analysed per microglia) appearance at phagosome, after its formation. Violin plots have mean and quartiles depicted. (**C**) Microglia (magenta; Tg(*csf1ra:GAL4-VP16; UAS:lyn-tagRFPt*)) and Rab5-positive endosomes (cyan*; UAS:mNeonGreen-Rab5a*), full time lapse in *Figure 6—video 1*. Endosomes form a cloud (arrowhead) that moves into phagocytic branches towards forming phagosomes (asterisk). Original images were deconvolved using Huygens deconvolution. (**D**) Representative microglia (magenta; Tg(*csf1ra:GAL4-VP16; UAS:lyn-tagRFPt*)), Rab5 endosomes (cyan; *UAS:mNeonGreen-Rab5a*), and centrosome (yellow;

*Figure 6 continued on next page*

*Figure 6 continued*

*UAS:miRFP670-centrin4*). The centrosome (circle) moves into phagocytic branches (star) and is followed by the cloud of Rab5 endosomes (arrowhead). Original images were deconvolved using Huygens deconvolution. n refers to the number of microglia examined.

The online version of this article includes the following video, source data, and figure supplement(s) for figure 6:

**Source data 1.** Related to *Figure 6B*.

**Figure supplement 1.** Vesicular markers do not affect microglia phagocytosis.

**Figure supplement 1—source data 1.** Related to *Figure 6—figure supplement 1B-C and E-F*.

**Figure 6—video 1.** Rab5-positive endosomal cloud is highly dynamic and moves into phagocytic branches.

https://elifesciences.org/articles/82094/figures#fig6video1

## The centrosome plays a key role in limiting the rate of neuronal engulfment in microglia

As microglia have only one centrosome, this cannot be in two branches simultaneously and it might limit the number of phagocytic events and the rate of neuronal engulfment. To directly test this hypothesis experimentally, we exploited the fact that increased *centrin 4* expression can generate microglia with more than one centrosome, which we refer to as 'double-centrosome microglia' (*Figure 7A*, *Video 7*). Moreover, mosaic expression of the Gal4/UAS system (*Halpern et al., 2008*) allowed the generation of double-centrosome microglia and matching controls with only one centrosome within the same brain (*Figure 7A*). Interestingly, we found that double-centrosome microglia showed significantly increased neuronal phagocytosis when compared to controls (*Figure 7B and C*, *Video 7*). Live imaging of double-centrosome microglia revealed that occasionally phagocytic events can also occur simultaneously at the tip of two different branches, with each branch having a centrosome in the vicinity of the newly formed phagosome (*Figure 7B*). Such concurrent phagocytic events are usually not observed in control microglia that have only one centrosome. Thus, we conclude that this organelle, acting as the main MTOC, limits the rate of neuronal phagocytosis in these cells.

## A DAG/PLC-dependent pathway participates in centrosomal reorientation and movement towards forming phagosomes

We next turned our attention to how the microglial centrosome is recruited towards forming phagosomes. The mechanism described here is reminiscent of the establishment of the IS between T cells and antigen-presenting cells (APCs), where T cells are known to polarize vesicular trafficking in a centrosomal-dependant manner (*Geiger et al., 1982*; *Martín-Cófreces et al., 2014*; *Stinchcombe et al., 2006*). In T cells, centrosomal recruitment is induced by the creation of a localized PLC-dependent DAG build-up at the IS (*Quann et al., 2009*). We took two complementary approaches to investigate whether this pathway also plays a role in centrosomal recruitment in microglia. We treated 3-dpf embryos with either phorbol 12-myristate 13-acetate (PMA), a well-known DAG analog that has been shown to mimic uniform DAG signalling (see schematic in *Figure 8A*; *Kochs et al., 1993*; *Nishizuka, 1992*; *Visnjić et al., 1995*), or with U-73122, a pan-inhibitor of PLC (PLCi; see schematic in *Figure 8A*; *Cheeseman et al., 2005*; *Quann et al., 2011*). Both treatments can be used to prevent localized DAG build-up and result in similar microglial phenotypes. Indeed, in both PLCi and PMA-treated embryos, the targeted movement of the centrosome into microglial branches is reduced (*Figure 8B*, *Video 8*), and this organelle can be found with a higher probability closer to the cell

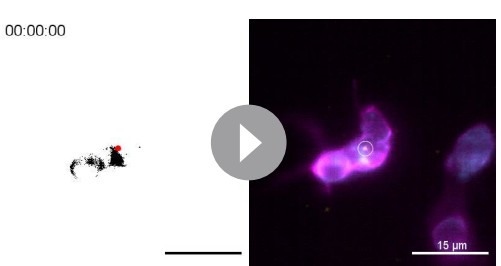

00:00:00

15 µm

**Video 6.** Rab5-positive early endosomes form a cloud that follows the centrosome into phagocytic branches. Right: representative microglia (magenta; Tg(*csf1ra:GAL4-VP16; UAS:lyn-tagRFPt*)), expressing the Rab5 endosome (cyan; *UAS:mNeonGreen-Rab5a*) and centrosome (yellow; *UAS:miRFP670-centrin4*) reporters. Left: image segmentation showing tracking of the centrosome (red dot) and a binary threshold view of the Rab5 signal (black pixels) to show cloud dynamics. Images were captured every 15 s for 30 min using single-plane illumination microscopy (SPIM). Timescale is hh:mm:ss.

https://elifesciences.org/articles/82094/figures#video6

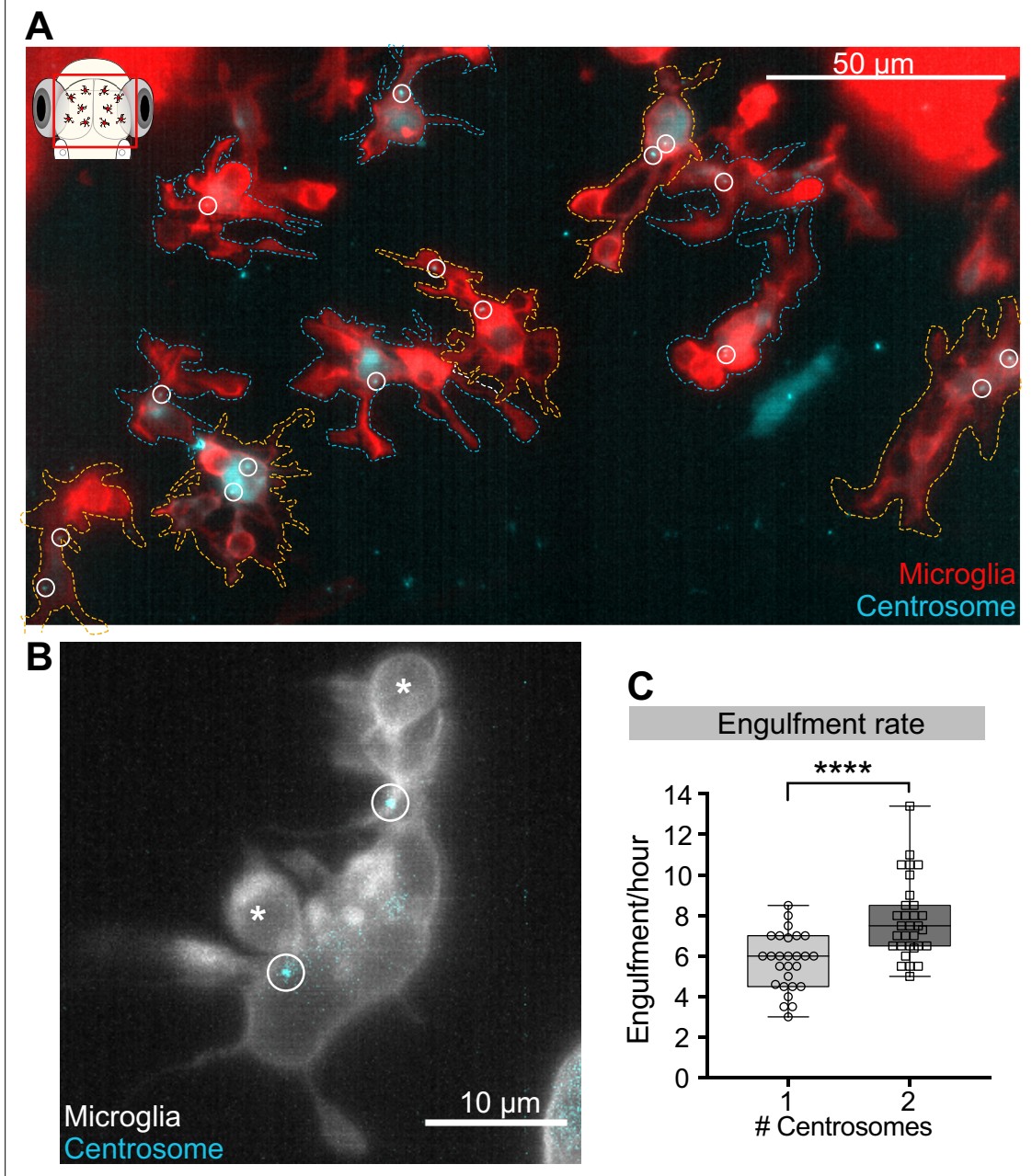

**Figure 7.** Microglia with two centrosomes are more efficient phagocytes. (**A**) Microglia (red; Tg(*csf1ra:GAL4-VP16; UAS:lyn-tagRFPt*)) overexpressing the centrosome marker (cyan; Tg(*UAS:miRFP670-cetn*4)). Due to mosaicism in the overexpression of *UAS:miRFP670-cetn4,* there are microglia (dashed outlines) with two centrosomes (orange cells) and matching controls with only one (blue cells) within the same brain. Centrosomes are labelled with circles. (**B**) Microglia (grey) overexpressing the centrosome marker (cyan) resulting in this cell having two centrosomes (circle). (**C**) Microglia with two centrosomes engulf more neurons (N = 8, 2–9 microglia analysed per fish). Boxplots depict mean +/- min and max values. Groups were compared using a two-tailed, nonparametric Mann-Whitney U test, ****p<0.0001. N refers to the number of zebrafish analysed.

The online version of this article includes the following source data for figure 7:

**Source data 1.** Related to *Figure 7C*.

centre (*Figure 8B and C*, *Figure 8—figure supplement 1*). This coincides with a decrease in successful engulfments (*Figure 8E*), despite microglia in treated embryos are highly dynamic (as shown for PMA in *Figure 8D* and *Video 8*) and contact the same number of apoptotic neurons as DMSO controls (*Figure 8F*). This indicates that microglia can still sense and contact apoptotic targets but cannot engulf, a phenotype that is highly reminiscent of aborted phagocytic attempts in which branches

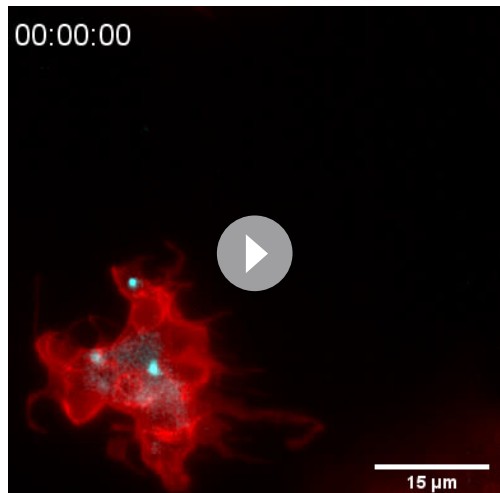

**Video 7.** Microglia with two centrosomes. Representative microglia (red; Tg(*csf1ra:GAL4-VP16; UAS:lyn-tagRFPt*)) overexpressing the centrosome reporter (cyan; Tg(*UAS:miRFP670-cetn4*)) resulting in two centrosomes. Images were captured every 20 s for 40 min using single-plane illumination microscopy (SPIM). Timescale is hh:mm:ss.

https://elifesciences.org/articles/82094/figures#video7

that do not recruit the centrosome fail to engulf (*Figure 4G and H*). Interestingly, we also found that in those fewer cases where microglia in PMA and PLCi-treated embryos manage to engulf, this always coincides with the orientation and movement of the centrosome towards successful phagosomes (*Figure 8G and H*).

Thus, experiments with these molecular inhibitors point to a role for the PLC/DAG signalling cascade in promoting long-range targeted movements of the centrosome towards forming phagosomes, reinforcing the link between centrosomal localization and successful neuronal engulfment in microglia. Furthermore, taken together these data highlight striking cellular and molecular similarities between the immunological and phagocytic synapse supporting a possible evolutionary link between these two structures.

## Discussion

While the importance of microglia as brain phagocytes is well recognized, less clear is how these cells identify and engulf neuronal targets effectively. Live imaging in various systems has shown that microglia have highly dynamic branches of different widths and lengths (*Bernier et al., 2019*). One hypothesis is that they use these processes to constantly scan the brain parenchyma to identify, for example, apoptotic neurons to engulf. Thus, it can be assumed that the more branches a microglial cell can generate, the more neuronal targets it can find and remove. However, observations in the zebrafish indicate that intrinsic mechanisms might be in place to regulate phagocytosis in microglia. Indeed, the engulfment rate was found to depend on cargo digestion and previous experiments have also shown that phagosome formation can be aborted (*Mazaheri et al., 2014*; *Villani et al., 2019*).

In this study, we show that microglia that have many branches and are surrounded by multiple apoptotic neurons engulf sequentially by selecting one branch at a time. This branch selection strongly correlates with the movement of the microglial centrosome that can be seen moving into one branch towards the forming phagosome. This migration coincides with successful neuronal engulfment and

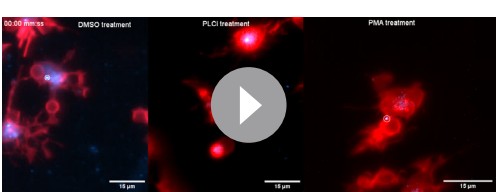

**Video 8.** Microglia centrosome is retained within the cell soma after phorbol 12-myristate 13-acetate (PMA) and pan-inhibitor of PLC (PLCi) treatment. Representative microglia treated with DMSO, PMA, and PLCi (grey; Tg(*csf1ra:GAL4-VP16; UAS:lyn-tagRFPt*)) expressing the centrosome reporter (cyan; Tg(*UAS:miRFP670-cetn4*)). Images were captured every 30 s for 1 hr using single-plane illumination microscopy (SPIM). Original images were processed with a Gaussian Blur filter with sigma(radius) = 1. Timescale is mm:ss.

https://elifesciences.org/articles/82094/figures#video8

polarized vesicular trafficking, pointing to the centrosome acting as dynamic cellular MTOC. Thus, the centrosome effectively also contributes to limiting engulfment in microglia and can be considered to act as a bottleneck in the removal of dying neurons. Indeed, the fact that microglia have only one centrosome per cell can explain why neuronal removal always occurs sequentially as this organelle cannot be in two places simultaneously. Indeed, we found that doubling the number of centrosomes in microglia increases the engulfment rate allowing these cells to engulf two neurons at the same time, indicating that the movement of this organelle is rate-limiting and sufficient to promote neuronal removal. Interestingly, we have found that the centrosome does not migrate towards aborted phagocytic attempts and we speculate that this could be due to low DAG signalling. In line with this, drug experiments

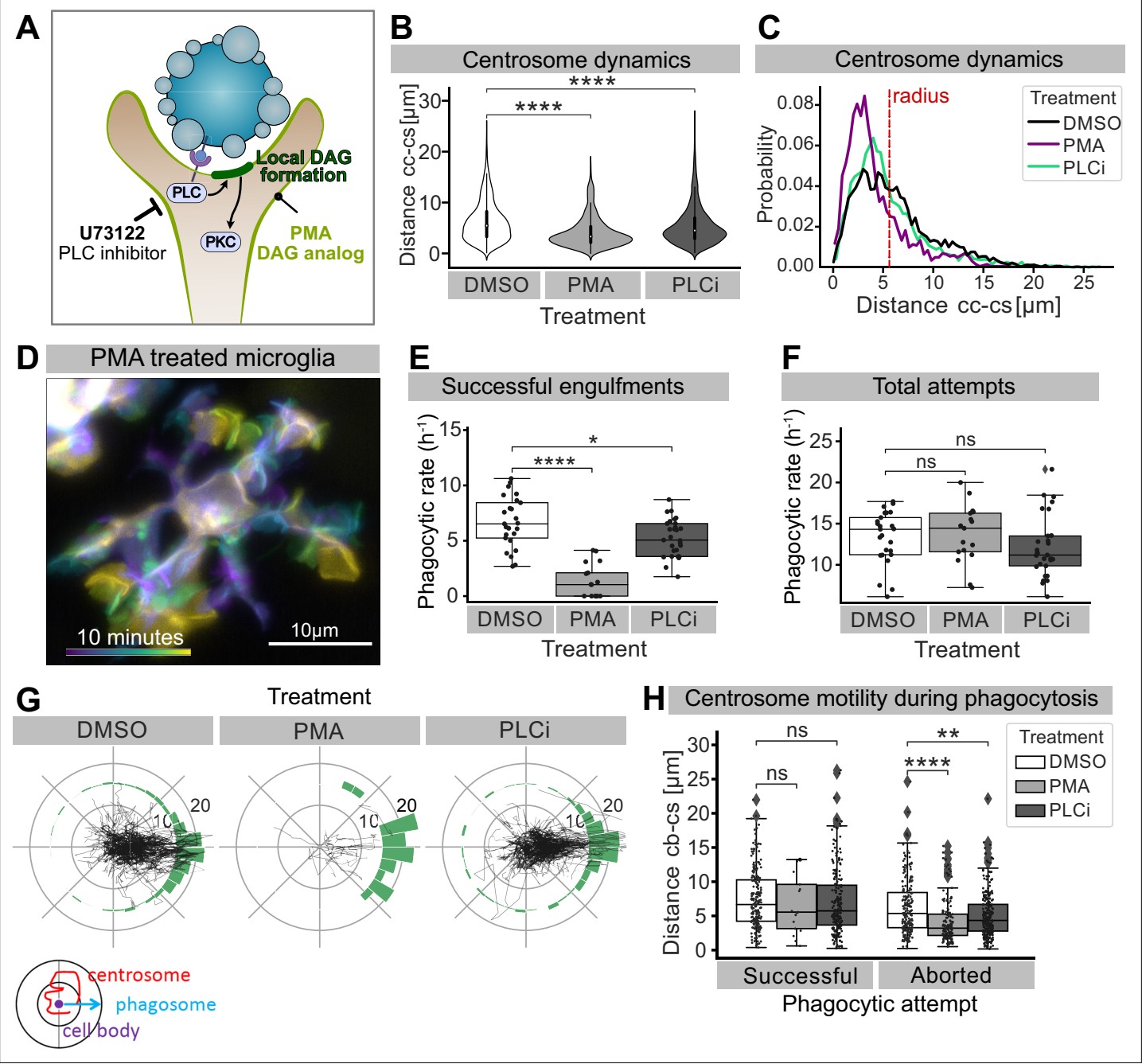

**Figure 8.** Perturbing DAG signalling affects centrosomal motility and phagocytosis. (**A**) Schematic showing the effect of phorbol 12-myristate 13-acetate (PMA) and pan-inhibitor of PLC (PLCi) on DAG-mediated signalling. (**B**) Distance of the centrosome from the cell centre in microglia after 1% DMSO (N = 4, n = 26), 2.5 µM PMA (N = 3, n = 17) or 1 µM U73122 (N = 4, n = 29) treatment. (**C**) Probability map of the radial location of the centrosome, relative to the cell centre, after DMSO, PMA, or PLCi treatment. (**D**) A colour-coded overlay of a 10 min time lapse of a representative microglia (Tg(*mpeg1:Gal4; UAS:lyn-tagRFPt*)) treated with 2.5 µM PMA, to show branch dynamics. (**E, F**) Phagocytic attempts by microglia treated with 1% DMSO (white), 2.5 µM PMA (light grey) or 1 µM U73122 (dark grey). Plots show (**E**) successful and (**F**) total phagocytic attempts (*p<0.05, ****p<0.0001). (**G**) Tracks of the centrosome relative to successful phagocytic events, after DMSO, PMA, or PLCi treatment. Histograms summarize the position of the centrosome at the time of phagosome formation. (**H**) Centrosome distance from cell centre at the time of successful or aborted phagocytic attempts, after DMSO, PMA, or PLCi treatment (*p<0.05, **p<0.01, ***p<0.001, ****p<0.0001). Violin and boxplots depict mean and 1.5x interquartile range. Groups were compared using an unpaired, nonparametric Kruskal-Wallis with Bonferroni correction. N refers to the number of zebrafish and n to the total number of microglia analysed.

The online version of this article includes the following source data and figure supplement(s) for figure 8:

*Figure 8 continued on next page*

*Figure 8 continued*

**Figure supplement 1.** The radius of microglia cell soma.

**Figure supplement 1—source data 1.** Related to *Figure 8—figure supplement 1*.

show that when DAG signalling is perturbed, the centrosome remains mostly within the cell soma and phagocytosis is reduced.

Microglial branches can be divided into two categories: thin actin-based filopodia that have been suggested to play a critical role in scanning the brain parenchyma, and thicker microtubule-based branches that have been shown to form a stable cellular backbone. Interestingly, in adult brains, these stable processes have been shown to move and extend toward neuronal injuries, indicating a potential role for these structures in recognizing dying neurons (*Bernier et al., 2019*; *Davalos et al., 2005*). Our study has found that in development microglial microtubule-based branches are highly dynamic and responsible for removing the numerous apoptotic neurons that are scattered within the brain. Light-mediated depolymerization of microtubules can be used to prevent these branches from forming. This dramatically impacts microglial morphology and neuronal engulfment, with these cells migrating directly towards dying neurons in a macrophage-like fashion. This highlights two important aspects: the first is that reducing microtubule stability results in microglia acquiring an amoeboid morphology and behaviour that could be considered typical of tissue macrophages, indicating that changes in microtubule organization might feedback on cell state, and the second interesting insight is the fact that microglia possess incredible robust plasticity in executing the removal of apoptotic neurons during brain development; they can branch or move directly towards these cells. Remarkably, both modes of engulfment are effective, raising the question of why BM engulfment predominates under normal circumstances. A potential reason for microglia preferring extensions instead of migrating directly towards corpses under physiological conditions could be the need to limit the disruption of the delicate brain parenchyma. Interestingly, experiments in *Drosophila* have shown that macrophages engulf apoptotic corpses by migrating directly towards these targets, but when the formation of branched actin networks is inhibited, they show impaired motility and utilize long extensions (*Davidson and Wood, 2020*). Thus, in both systems perturbing normal cytoskeletal organization in professional phagocytes appear to feedback on cell state pushing these cells to adopt alternative modes of engulfment, further emphasizing how incredibly robust efferocytosis can be.

The contact between the phagocyte and the apoptotic cell has been named the phagocytic – or efferocytotic – synapse (*Barth et al., 2017*; *Kelley and Ravichandran, 2021*), and this structure has been suggested to share important similarities with another critical cell–cell interphase, the immunological synapse, between T cells and APCs (*Dustin, 2012*; *Niedergang et al., 2016*). Indeed, these are both characterized by clustering of receptors and targeted vesicular trafficking (*Niedergang et al., 2016*). Our investigation into how microglia engulf dying neurons reveals additional important parallels between the two. The first is the reorientation of the centrosome that is linked to cytoskeletal polarization and vesicular trafficking, and the second is that in both this movement depends on the PLC/DAG signalling cascade. These data make a compelling case for the existence of evolutionary conservation between these two synapses. A better understanding of their shared features could help to develop new therapeutic strategies to target both immunological and phagocytic disorders.

## Materials and methods

**Key resources table**

| Reagent type (species) or resource | Designation | Source or reference | Identifiers | Additional information |
|---|---|---|---|---|
| Genetic reagent (*Danio rerio*) | Tg(mpeg1.1:EGFP-caax) | *Villani et al., 2019* | ZFIN ID: ZDB-TGCONSTRCT-191211-1 | Macrophage membrane labelling |
| Genetic reagent (*D. rerio*) | Tg(mpeg1.1:Gal4-VP16) | *Ellett et al., 2011* | ZFIN ID: ZDB-ALT-120117-3 | Macrophage Gal4 driver line |
| Genetic reagent (*D. rerio*) | TgBAC(csf1ra:Gal4-VP16)i186 | *Gray et al., 2011* | ZFIN ID: ZDB-ALT-110707-2 | Macrophage Gal4 driver line |

*Continued on next page*

*Continued*

| Reagent type (species) or resource | Designation | Source or reference | Identifiers | Additional information |
|---|---|---|---|---|
| Genetic reagent (*D. rerio*) | Tg(Xla.Tubb:LEXPR-SEC-Hsa. ANXA5-TagBFP) | *Mazaheri et al., 2014* | ZFIN ID: ZDB-ALT-170110-1 | Reporter line for apoptotic neurons |
| Genetic reagent (*D. rerio*) | Tg(UAS:EMTB-3xGFP) | *Revenu et al., 2014* | | UAS-driven microtubule reporter |
| Genetic reagent (*D. rerio*) | Tg(UAS:EB3-GFP) | *Fassier et al., 2018* | | UAS-driven microtubule reporter |
| Genetic reagent (*D. rerio*) | Tg(UAS:mNeonGreen-UtrCH) | *Shkarina et al., 2021* | | UAS-driven actin reporter |
| Genetic reagent (*D. rerio*) | Tg(UAS:lyn-tagRFPT) | This paper | | UAS-driven red membrane labelling |
| Genetic reagent (*D. rerio*) | Tg(UAS:mNeonGreen-Rab5a) | This paper | | UAS-driven early endosomal reporter |
| Genetic reagent (*D. rerio*) | Tg(UAS:miRFP670-cetn4) | This paper | | UAS-driven centrosomal reporter |
| Genetic reagent (*D. rerio*) | Tg(UAS:Lamp1-mGFP) | This paper | | UAS-driven lysosomal reporter |
| Genetic reagent (*D. rerio*) | Tg(UAS:EB3-mScarlet-I) | This paper | | UAS-driven microtubule reporter |
| Recombinant DNA reagent | UAS:EB3-mScarlet-I (plasmid) | This paper | RRID:Addgene_192358 | Made with tol2kit, see section 'Cloning of plasmids' |
| Sequence-based reagent | Sp6-EB3-mScarlet_Fw | This paper | PCR primers | ATTTAGGTGACACTAT AGAAGATGGCCGTC AATGTGTAC |
| Sequence-based reagent | Sp6-EB3-mScarlet_Rev | This paper | PCR primers | CAACTATGTATA ATAAAGTTG |
| Commercial assay or kit | MultiSite Gateway Pro | Invitrogen | #12537 | |
| Commercial assay or kit | Gibson Assembly Master Mix | New England Biolabs | #E2611 | |
| Commercial assay or kit | pENTR/D-TOPO Cloning Kit | Invitrogen | # K240020SP | |
| Commercial assay or kit | mMESSAGE mMACHINE SP6 Transcription Kit | Invitrogen | #AM1340 | |
| Commercial assay or kit | MinElute Gel extraction kit | QIAGEN | #28604 | |
| Commercial assay or kit | GeneJet RNA Cleanup and Concentration kit | Thermo Fisher Scientific | #K0841 | |
| Chemical compound, drug | N-Phenylthiourea (PTU) | Sigma-Aldrich | #P7629 | |
| Chemical compound, drug | Tricaine (Mesab) | Sigma-Aldrich | #A5040 | |
| Chemical compound, drug | DMSO | Sigma-Aldrich | #D8418 | |
| Chemical compound, drug | Nocodazole | Sigma-Aldrich | #M1404 | |
| Chemical compound, drug | Photostatin-1 (PST-1) | *Borowiak et al., 2015* | | |
| Chemical compound, drug | Phorbol 12-myristate 13-acetate (PMA) | Sigma-Aldrich | #P8139 | |

*Continued on next page*

*Continued*

| Reagent type (species) or resource | Designation | Source or reference | Identifiers | Additional information |
|---|---|---|---|---|
| Chemical compound, drug | U73122 (PLC inhibitor) | Tocris | #U73122 | |
| Software, algorithm | Prism 9 | GraphPad | RRID:SCR_002798 | |
| Software, algorithm | Python 3.7 | Python Software Foundation | RRID:SCR_008394 | https://www.python.org/psf/ |
| Software, algorithm | Matplotlib 3.4.1 | *Hunter, 2007* | RRID:SCR_008624 | |
| Software, algorithm | NumPy 1.20 | *Harris et al., 2020* | RRID:SCR_008633 | |
| Software, algorithm | SciPy 1.4 | *Virtanen et al., 2020* | RRID:SCR_008058 | |
| Software, algorithm | Pandas 1.2.4 | *McKinney, 2010* | RRID:SCR_018214 | |
| Software, algorithm | Seaborn 0.11.1 | *Waskom, 2021* | RRID:SCR_018132 | |
| Software, algorithm | Imaris 9.5 | Bitplane | RRID:SCR_007370 | https://imaris.oxinst.com/ |
| Software, algorithm | Fiji | *Schindelin et al., 2012* | RRID:SCR_002285 | imagej.net/Fiji |
| Software, algorithm | BigDataProcessor2 (ImageJ plugin) | *Tischer et al., 2021* | https://github.com/bigdataprocessor/bigdataprocessor2 | |
| Software, algorithm | MtrackJ (ImageJ plugin) | *Meijering et al., 2012* | https://github.com/imagescience/MTrackJ/ | |
| Software, algorithm | HyperStackReg (ImageJ plugin) | *Sharma, 2018* | https://github.com/ved-sharma/HyperStackReg/tree/v5.6 | |
| Software, algorithm | Z-stackDepthColorCode (ImageJ plugin) | | https://sites.imagej.net/Ekatrukha/ | |
| Software, algorithm | Analysis code | Max Brambach | https://github.com/max-brambach/microglia_centrosome | |

## Animal handling

Zebrafish (*Danio rerio*) were raised, maintained, and bred according to standard procedures as described in "Zebrafish – A practical approach" (*Nüsslein-Volhard, 2012*). All experiments were performed on embryos younger than 4 dpf, in accordance with the European Union Directive 2010/62/EU and local authorities (Kantonales Veterinämt; Fishroom licence TVHa Nr. 178). Sex determination is not possible since zebrafish only become sexually different around 20–25 dpf. Live embryos were kept in E3 buffer at 28–30°C, and staging was done according to *Kimmel et al., 1995*. Pigmentation was prevented during experiments by treating the embryos with 0.002% N-phenylthiourea (PTU) (Sigma-Aldrich, St. Louis, MO, #P7629) from 1 dpf onwards. Embryos were anaesthetized during mounting procedures and experiments using 0.01% tricaine (Sigma-Aldrich, #A5040).

## Transgenic fish lines

The following transgenic animals were used in this study: Tg(*mpeg1:GFP-caax*) (*Villani et al., 2019*), Tg(*nbt:dLexPR-LexOP:secA5-BFP*) (*Mazaheri et al., 2014*), Tg(*mpeg1:Gal4, UAS:Kaede*) (*Ellett et al., 2011*; *Sieger et al., 2012*), or TgBAC(*csf1ra:GAL4-VP16; UAS-E1B:NTR-mCherry*) (*Gray et al., 2011*). A new membrane marker Tg(*UAS:lyn-tagRFPt*) was generated during this study using MultiSite Gateway Pro cloning (Invitrogen, Waltham, MA) and the Tol2 kit (*Kwan et al., 2007*), and crossed with

either of the Gal4 driver lines to visualize microglia: Tg(*mpeg1:Gal4, UAS:lyn-tagRFPt*) or TgBAC(*csf1ra:GAL4-VP16; UAS:lyn-tagRFPt*). Both combinations are referred to as 'membrane line' but the exact genotype for each experiment is indicated in the figure legends. The following Upstream Activation Site (UAS) reporter lines were crossed with the Gal4 driver lines or the membrane lines to visualize expression for actin: Tg(*UAS:mNeonGreen-UtrCH*) (*Shkarina et al., 2021*) and microtubules: Tg(*UAS:EMTB-3xGFP*) (*Revenu et al., 2014*) and Tg(*UAS:EB3-GFP*) (*Fassier et al., 2018*). Furthermore, the following UAS lines were generated using Gateway cloning and the Tol2 kit and crossed with the membrane lines to visualize expression for the centrosome: Tg(*UAS:miRFP670-cetn4*), early endosomes: Tg(*UAS:mNeonGreen-Rab5a*), lysosomes: Tg(*UAS:Lamp1-mGFP*), and microtubules: Tg(*UAS:EB3-mScarlet-i*). In one subset of the data, an *EB3-mScarlet-i* was injected as mRNA.

## Cloning of plasmids

The following plasmids were generated by MultiSite Gateway Pro cloning strategy based on the Tol2kit (*Kwan et al., 2007*) using the Gateway LR Clonase II mix (Invitrogen #11791020).

| Final vector | p5E vector | pME vector | p3E vector | Transgenic marker |
| --- | --- | --- | --- | --- |
| pDEST_6xUAS:lyn-tagRFPt (RRID:Addgene_192354) | p5E_6xUAS | pME_lyn | p3E_tagRFPt | *cry:ECFP* |
| pDEST_6xUAS:mNeonGreen-Rab5a (RRID:Addgene_192355) | p5E_6xUAS | pME_mNeonGreen (*Shkarina et al., 2021*) | p3E_Rab5 (*Hartmann et al., 2020*) | *cmlc2:tagRFP* |
| pDEST_6xUAS:Lamp1-mGFP (RRID:Addgene_192356) | p5E_6xUAS | pME_Lamp1-mGFP | p3E_polyA | *cmlc2:tagRFP* |
| pDEST_6xUAS:miRFP670-cetn4 (RRID:Addgene_192357) | p5E_6xUAS | pME_miRFP670 | p3E_cetn4 (*Revenu et al., 2014*) | *cry:mKate2* |
| pDEST_5xUAS:EB3-mScarlet-I (RRID:Addgene_192358) | p5E_6xUAS | pME_EB3-mScarlet-I | p3E_polyA | *cry:ECFP* |

The pME_miRFP670 was cloned using Gateway BP cloning using pmiRFP670-N1, a gift from Vladislav Verkhusha (Addgene #79987; *Shcherbakova et al., 2016*), as a template. The pME_Lamp1-mGFP vector was cloned using the pENTR/D-TOPO Cloning kit (Invitrogen #K240020) according to the manufacturer's instructions, using Lamp1-mGFP, a gift from Esteban Dell'Angelica (Addgene plasmid #34831; *Falcón-Pérez et al., 2005*) as a template. The pME_EB3-mScarlet-I was cloned using the Gibson Assembly cloning kit (NEB #E2611S) according to the manufacturer's instruction. As templates for the assembly, the pME_mNeonGreen was used as the backbone while pC2_EB3-GFP (*Revenu et al., 2014*) and pmScarlet-i_C1, which was a gift from Dorus Gadella (Addgene plasmid #85044; *Bindels et al., 2016*), were templates for the inserts. The following primers were used for the cloning:

| Vector | Primer sequence (5'–3') |
| --- | --- |
| Lamp1-mGFP pENTR/D-TOPO cloning | FW CACCTTCAGGGACATGGCGGCC<br>REV TTACTTGTACAGCTCGTCCATGCCG |
| pC2_EB3-GFP Gibson Assembly P1 | FW CAGGCTGGATGGCCGTCAATGTGTAC<br>REV GGTGGCGACCGGTGGATCCAGGTACTCGTCCTGGTCTTC |
| pmScarlet-i_C1 Gibson Assembly P2 | FW CTGGATCCACCGGTCGCCACCATGGTGAGCAAGGGCGAG<br>REV TTACTTGTACAGCTCGTCCATGC |
| pME_mNeonGreen Gibson Assembly P3 | FW GACGAGCTGTACAAGTAAGACCCAGCTTTCTTGTAC<br>REV CGGCCATCCAGCCTGCTTTTTTGTAC |
| p3E_cetn4 Gateway BP cloning | FW GGGGACAGCTTTCTTGTACAAAGTGGGGatggcgtccggcttcaggaaaag<br>REV GGGGACAACTTTGTATAATAAAGTTGGgtacagattggtttcttcataat |
| pME_miRFP670 Gateway BP cloning | FW GGGGACAAGTTTGTACAAAAAAGCAGGCTGGATGGTAGCAGGT CATGCCTCTGgcagccc<br>REV GGGGACCACTTTGTACAAGAAAGCTGGGTCGCTCTCAAGCGCG GTGAtccgc |

All plasmids generated during this study can be requested and will be made available in Addgene.

## mRNA synthesis

The pDEST_5xUAS:EB3-mScarlet-I plasmid was used as a template to generate the *EB3-mScarlet-I* mRNA. The following primers were used to amplify the sequence:

| Primer | Sequence (5'->3') |
| --- | --- |
| Sp6-EB3-mScarlet-Fw | ATTTAGGTGACACTATAGAAGATGGCCGTCAATGTGTAC |
| Sp6-EB3-mScarlet-Rev | CAACTATGTATAATAAAGTTG |

The amplicon was purified using the QIAGEN MinElute Gel extraction kit (QIAGEN #28604). Synthesis of *EB3-mScarlet-I* mRNA was carried out using the mMessage mMachine Sp6 kit (Thermo Fisher #AM1340) following the manufacturer's instructions. mRNA was recovered using the GeneJet RNA Cleanup and Concentration kit (Thermo Fisher #K0841). Purified mRNA was diluted to 250 ng/µl and stored in 4 µl aliquots at –80°C.

## Chemical perturbations

### Photostatin treatments

*For validation of PST-1 functionality in zebrafish embryos*: Eggs at the 1–2 cell stage were injected with ~2 nl *EB3-mScarlet-I* mRNA (250 ng/µl). At 4 hr post-fertilization (hpf) and 24 hpf, embryos that showed expression of the transgene were treated with either 1% DMSO or 1 µM PST-1 in 1% DMSO for 1 hr and kept in the dark. They were then dechorionated and mounted while covering the light source with orange plexiglass to prevent spontaneous activation.

*For experiments on microglia at 3 dpf*: Embryos were preincubated in 1% DMSO or 20 µM PST-1 in 1% DMSO for 1 hr and kept in the dark before they were mounted and imaged. They were imaged with 561 nm light for another hour before 'activating' the drug, which was achieved by illuminating the embryos with 405 nm light between each time frame. For 'inactivating' the drug again, the embryos were imaged as before with 561 nm laser light and illuminated with 488 nm light between each time frame.

### Nocodazole treatment

Embryos were preincubated for 1 hr in 1% DMSO or 10 µM nocodazole (Sigma-Aldrich) in 1% DMSO. The same treatment was maintained during imaging.

### PLCi and PMA treatment

Embryos were preincubated in 1% DMSO for 2–3 hr. *For PLCi*: Embryos were incubated in 1 µM U73122 (Tocris, Bristol, UK) and 1% DMSO for another 35 min before moving them to the light-sheet microscope chamber, where they were imaged in medium containing 0.5 µM U73122 and 1% DMSO. *For PMA treatment*: Embryos were mounted in the light-sheet chamber and kept in 1% DMSO until right before imaging started, then the chamber medium was replaced with E3 containing 2.5 µM PMA and 1% DMSO.

## High-resolution live imaging

Embryos were anaesthetized in 0.01% tricaine and pre-selected based on the expression of the desired fluorophore using a Nikon ZMZ18 fluorescent stereoscope. Embryos were embedded in 1–1.2% low-melting (LM) agarose (PeqGOLD Low Melt Agarose, PeqLab Biotechnologie GmbH), dissolved in E3 medium with 0.01% tricaine. Embryos were mounted on glass-bottom dishes (Greiner Bio-One #627871) for confocal microscopy or pulled together with agar into glass capillaries (Brand, #701904) with a rod (Brand, #701932), and then pushed halfway out, into the microscopy chamber for light-sheet microscopy. Both the imaging dishes and the microscopy chamber were filled with E3 medium containing 0.002% PTU and 0.01% tricaine during the entire imaging period. For confocal microscopy in *Figure 6—figure supplement 1*, Leica SP8 Inverted microscope with a Leica ×40/NA 1.1 WI objective was used to capture 75-µm-thick stacks with z-step of 1 µm. For all other confocal microscopy, an Andor Dragonfly 200 Sona spinning-disc microscope with a Nikon ×20/NA 0.95 WI objective and

40 µm spinning disc was used to capture imaging 70–75-µm-thick stacks with z-steps of 2 µm, every 60 s. For light-sheet microscopy, a Zeiss Z.1 microscope with a W Plan-APO ×20/NA 1.0 WI imaging objective and ×10/NA 0.2 air illumination objectives was used to capture 40–75-µm-thick stacks with a z-step of 0.5–0.75 µm, every 15–30 s.

## Light-sheet image pre-processing

For 3D analysis of light-sheet microscopy images, 3D volumes were converted to H5 files using Fiji BigDataProcessor (*Tischer et al., 2021*). Cells were tracked and individual frames were saved using BigDataTools and BigDataTracker for further analysis. For 2D analysis of light-sheet microscopy images, maximum intensity projections were used and drift corrected when necessary using the Fiji HyperStackReg plugin (*Sharma, 2018*).

## Image analysis

### Microglia and neuronal cell numbers and nearest-neighbour analysis

Imaris automatic spot detection with manual corrections was used to detect individual microglia and dead neurons from spinning disc images. The coordinates of the microglia were exported and used to calculate their nearest neighbour using k nearest-neighbour search (k = 1; scipy.spatial.KDTree) (*Virtanen et al., 2020*).

### Microglia mobility

Imaris automatic spot detection with manual corrections was used to place and track microglia from time-lapse recordings acquired with a spinning disc microscope. The total displacement along the track divided by the total tracking time yielded the mean speed of the microglia.

The persistence of microglia migration was quantified using the mean square displacement $\alpha$, a metric that describes how 'random' the motion of a cell is. For that, the coordinate vectors of the microglia $x_i$ along their track were used to calculate the mean square displacement (a measure of the area/volume explored by a randomly walking cell during a given time interval):

$$\xi\left(n\right) = \frac{1}{N-n+1} \sum_{i=0}^{N-n} \delta'_{i+n,i} \delta_{i+n,i}$$

with the total number of time steps of a track $N$, the step size $n$, and the displacement vector between two positions on the track $\delta_{a,b} = x_a - x_b$. $\xi$ was calculated for all integer $n \in \left[1, N\right]$. Subsequently, linear regression on $log\left(\xi\left(n\right)\right)$ $log\left(n\right)$ yielded the slope $\alpha$.

For $0 < \alpha$, migration can be considered subdiffusive (i.e. spatially constrained diffusion), $\alpha = 0$ indicates regular diffusion (i.e. random walk), $1 < \alpha < 2$ indicates superdiffusion (i.e. superimposition of diffusion and directional movement), and $\alpha = 2$ for straight tracks (*Gorelik and Gautreau, 2014*; *Metzler and Klafter, 2004*).

### Microglia phagocytosis

Maximum intensity projections were used to count phagocytic events using the Fiji plugin Cell Counter or MTrackJ. For successful events, a dot was added to the first time point after phagosomal closure. For aborted attempts, a dot was added to the second time point of phagocytic cup formation. To count the phagocytic rate, the total number of counts was divided by the duration of the time lapse for each cell.

### Microglia branch dynamics

The length of phagocytic branches was obtained by measuring the distance between the centre of the cell body and the phagosome, at the first time point of the phagosome track.

For branch dynamics, individual 3D cell volumes, acquired via light-sheet microscopy, were used. Imaris manual spot detection was used to track tips of either filopodia or large cellular extensions of microglia from time-lapse recordings acquired with light-sheet microscopy. The total displacement along the tracks divided by the total tracking time yielded the mean speed of the branch extensions and retractions.

## Microglia segmentation and sphericity measurement

Imaris surface detection was used to segment individual microglia over time from spinning disc images in *Figure 3*. Volume $V$ and surface area $A$ of the microglia were exported and used to calculate the sphericity $\phi$ via

$$\Psi = \pi^{\frac{1}{3}} \frac{(6V)^{\frac{2}{3}}}{A}$$

## Quantifications of endosomes and lysosomes

We took advantage of the chimeric expression of the Rab5 and Lamp1 reporters to compare microglia that express either both cytoplasmic maker and vesicular marker, or only the cytoplasmic marker, within the same brain. The analysis was performed in a blinded manner; only the mCherry channel was used for the measurements, and only afterwards was the Rab5 or Lamp1 channel visualized to determine which microglia expressed the reporter. Measurements were performed on the 3D stacks in Fiji by moving through the image planes and finding the biggest diameter of each vesicle. The 'line' tool was used to draw a straight line across the vesicle. If a vesicle was elliptical in shape, the diameter was measured approximately where it was halfway between biggest and smallest. The measurement tool in Fiji was used to extract both the number and size of the vesicles.

## Centrosome dynamics

For the analysis of centrosome dynamics, cropped time-lapse data of individual microglia was used to track the cell centre (cc), centrosome (cs), and phagosome (p) over time.

For the characterization of the wildtype microglia centrosome dynamics, the tracking was performed in 3D using Imaris automatic spot detection with manual corrections (cs, cc) and manual spot detection (p). For the drug treatment comparison, the tracking was achieved on 2D maximum intensity projections of volumetric light-sheet imaging data using the MTrackJ plugin of Fiji. Phagocytosis events were classified as 'successful' if a phagocytic cup closed and was transported into the cell soma, or as 'aborted' if a phagocytic cup existed for 1.5 min or longer but failed to close or was released again after some time. For 3D tracking, the phagosomes were marked in their centre and tracked subsequently to full closure until entering the cell soma; the aborted phagocytic attempts were marked at the cup 30 s after initial contact and tracked for three time points. For 2D tracking, the phagosomes were marked in their centre upon closure but not further tracked. The aborted phagocytic attempts were marked at the cup 30 s after initial contact.

To quantify the dynamic motion of the cs relative to cc and p, the triangle cc-cs-p was analysed. The coordinates for cs and cc were available for all time points of the tracks. The coordinates of p were only available from the formation of a phagosome on. Therefore, the temporally closest known location of the phagosome was used in cases where no tracking information was available. Temporal phagocytosis windows were defined as all time points closest to an engulfment attempt (successful or aborted) up to a maximum of ±10 min. The phagocytosis windows were aligned on the time point of the engulfment attempt and averaged to calculate the 'typical' centrosomal dynamics during phagocytosis.

To estimate the alignment of the cc-cs vector with the cc-p vector, the angle between the two was measured for individual phagocytosis events and the distribution of these angles at 1 min past the time of phagocytosis onset was recorded for every cell. Subsequently, the circular variance of these distributions was estimated using the approach of *Jammalamadaka and SenGupta, 2001* implemented in astropy.stats.circvar. A circular variance of 0 points towards a homogenous angular distribution on $(0, 2\pi)$ while a circular variance of 1 indicates that all angles are identical.

## Microglia radius estimates

The maximum intensity projections of light-sheet images of DMSO-treated embryos from *Figure 8* were used for this analysis. The oval tool in Fiji was used to make an oval, shaped as close to the shape of each microglia cell soma as possible. The measurement tool in Fiji was used to extract the diameter of the oval to calculate the radius.

## Statistical testing

Generally, unless otherwise specified, N refers to the number of zebrafish and n to the number of microglia analysed.

For statistical analysis, Prism 9 (GraphPad) and Python 3.7 were used. Unless otherwise specified, conditions were compared using an unpaired, two-tailed, nonparametric Mann–Whitney $U$-test. For comparing effects of microtubule polymerisation using PST-1 after turning the drug ON and OFF again, a nonparametric Friedmann test with a Dunn's correction was used. For comparing the effects of PMA and PLCi to DMSO controls, an unpaired, nonparametric Kruskal–Wallis with Bonferroni correction was used. p-Values were estimated in the following manner: *$0.05 > p > 0.01$, **$0.01 > p > 0.001$, ***$0.001 > p > 0.0001$, ****$p < 0.0001$.

## Acknowledgements

We are grateful to Oliver Thorn-Seshold for providing the Photostatin-1 and his help with using this in our system; to Jonas Hartmann for cloning the pDEST_UAS:lyn-tagRFPt construct; to Alexandra von Faber-Castell and Laura Comi for their help with analysing the vesicular markers; and to Cornelia Henkel for the fish care. Imaging was performed with support of the Center for Microscopy and Image Analysis, University of Zürich. This work was supported by a Swiss National Science Fund, Switzerland, grant no. 31003 A_182733 to FP and grant no. 310030_204834 to MB and DG. KM and FP designed the study. KM and AV conducted the experiments. KM, MB, and AV performed the data analysis. KM and EG generated the transgenic fishlines. FP and KM wrote the manuscript.

## Additional information

### Funding

| Funder | Grant reference number | Author |
|---|---|---|
| Schweizerischer Nationalfonds zur Förderung der Wissenschaftlichen Forschung | 31003A_182733 | Francesca Peri |
| Schweizerischer Nationalfonds zur Förderung der Wissenschaftlichen Forschung | 310030_204834 | Max Brambach Darren Gilmour |

The funders had no role in study design, data collection and interpretation, or the decision to submit the work for publication.

### Author contributions

Katrin Möller, Conceptualization, Resources, Data curation, Formal analysis, Validation, Investigation, Visualization, Methodology, Writing - original draft, Writing – review and editing; Max Brambach, Conceptualization, Software, Formal analysis, Validation, Investigation, Visualization, Writing – review and editing; Ambra Villani, Conceptualization, Data curation, Formal analysis, Validation, Investigation, Writing – review and editing; Elisa Gallo, Resources, Data curation; Darren Gilmour, Resources, Supervision, Funding acquisition, Writing – review and editing; Francesca Peri, Conceptualization, Supervision, Funding acquisition, Writing - original draft, Project administration, Writing – review and editing

### Author ORCIDs

Katrin Möller http://orcid.org/0000-0002-4720-8568
Max Brambach http://orcid.org/0000-0001-6194-9568
Ambra Villani http://orcid.org/0000-0002-4468-6353
Elisa Gallo http://orcid.org/0000-0003-2203-6787
Darren Gilmour http://orcid.org/0000-0001-7613-090X
Francesca Peri http://orcid.org/0000-0001-7068-4952

Decision letter and Author response
Decision letter https://doi.org/10.7554/eLife.82094.sa1
Author response https://doi.org/10.7554/eLife.82094.sa2

## Additional files

### Supplementary files
• MDAR checklist

### Data availability
All raw and processed imaging data produced in this study are openly available via the BioImageArchive (https://www.ebi.ac.uk/bioimage-archive/) under accession number S-BIAD564. All plasmids produced in this study have been submitted to Addgene. Numerical data obtained from these images are either provided as source data files or have been made available together with their respective analysis code on GitHub (https://github.com/max-brambach/microglia_centrosome, copy archived at swh:1:rev:b588b46398af76b96a3a69bb59b8e64852adda6c).

The following dataset was generated:

| Author(s) | Year | Dataset title | Dataset URL | Database and Identifier |
|---|---|---|---|---|
| Möller K, Villani A | 2022 | In vivo SPIM and confocal imaging of neuronal efferocytosis by microglia | https://www.ebi.ac.uk/biostudies/BioImages/studies/S-BIAD564?query=S-BIAD564 | BioImageArchive, S-BIAD564 |

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
