## [Editor Report]

This article is an important contribution to the microglia field and will be of interest to a broad readership in the fields of neurobiology, cell biology, and immunology. This work describes fundamental mechanisms of efferocytosis by microglia and uses impressive imaging in zebrafish, in combination with molecular manipulations, to provide compelling data of how centrosome movements synchronize with phagocytic cup formation during microglial efferocytosis of neuronal corpses in vivo.

---

## [Decision Letter]

**Decision letter after peer review:**

Thank you for submitting your article "A role for the centrosome in regulating the rate of neuronal efferocytosis by microglia in vivo" for consideration by *eLife*. Your article has been reviewed by 3 peer reviewers, and the evaluation has been overseen by a Reviewing Editor and Anna Akhmanova as the Senior Editor. The following individuals involved in review of your submission have agreed to reveal their identity: Cody J Smith (Reviewer #1); Eva Kiermaier (Reviewer #2).

The reviewers have discussed their reviews with one another, and the Reviewing Editor has drafted this to help you prepare a revised submission. All reviewers thought that this is an excellent paper. They have listed some suggestions which we ask you to consider, however, none of these were regarded as essential. When you resubmit the paper, please include a point-by-point response explaining the changes you made.

Essential revisions:

The Reviewers comments and suggestions are included for the authors to consider in their revised application. None of these were thought to be truly essential, however, and the reviewers leave it to the authors to decide how to modify their manuscript after reading the suggestions.

*Reviewer #1 (Recommendations for the authors):*

The presentation of the data is very clear and could be published as is.

*Reviewer #2 (Recommendations for the authors):*

This work is of high interest to a broad community. The data are well presented. Only a few aspects concerning data analysis need to be clarified and extended.

1. Figure 3J: what do the authors mean with MSDalpha? Why not show the MSD? The Figure is a bit confusing and needs more explanation/better presentation.

2. Figure 4E: the centrosome reporter seems to give a quite high background (derived from other cells?). At the very first picture many cyan dots can be seen. How did the authors identify the centrosome in microglia?

3. Figure 5: quantification of EB3/CETN4 migration events are missing.

4. Figure 5B: higher magnification picture would help to see MTs growing in the direction of phagosomes.

5. Figure 7: the authors claim that microglia with two centrosomes are more efficient phagocytes. They quantify the engulfment rate, which is significantly higher in the presence of two centrosomes. This looks very convincing. In the text they state that 'phagocytic events occurring simultaneously at the tip of two different branches, with each branch having a centrosome in the vicinity of the newly formed phagosome'. It would be nice to quantify the number of events, which show two phagocytic events simultaneously vs. consecutive phagocytic events.

*Reviewer #3 (Recommendations for the authors):*

Figure 1: Can the authors comment on what types of neurons are being engulfed? PVN, interneurons, etc?

Figure 4: A further discussion of the so-called 'aborted' phagocytic attempts would be useful. Is this a type of surveillance? Do they abort because the centrosome did not move, or does the centrosome only move when it senses an 'I'm really dead' signal from the neuron?

Figure 8: Representative images are lacking for PMA and PLCi treatment. What happens to microtubules and centrosome localization in these phenotypes? Figure 8D is only the mpeg reporter.

Discussion: I'm not a huge fan of calling many different kinds of things a synapse. 'immunologic synapse is by now a well used term, but a 'phagocytic synapse' feels like a step too far, especially for the neuroscience community.

---

## [Author Response]

Reviewer #2 (Recommendations for the authors):This work is of high interest to a broad community. The data are well presented. Only a few aspects concerning data analysis need to be clarified and extended.1. Figure 3J: what do the authors mean with MSDalpha? Why not show the MSD? The Figure is a bit confusing and needs more explanation/better presentation.

We are happy to clarify this point. The mean square displacement (MSD) of a cell indicates the area/volume explored by a randomly walking cell during a given time interval. The MSDalpha value is a metric that describes migration persistence, i.e. how ‘random’ the motion of a cell is. We derive this as follows: As time progresses, for a randomly moving cell, the MSD will increase. This relationship follows a power law of the form x(t) = b*t**a, in which the exponent a (or α) is determined by the directional persistence of the cell’s motion (a<1 confined, a=1 diffusion, a>1 directed, a=2 straight motion). We obtain this value by performing linear regression on log(x)~log(t).

Thus the MSDalpha value better describes if microglia migrate more directedly towards their targets upon microtubule destabilisation.

References:

Gorelik, R., Gautreau, A. Quantitative and unbiased analysis of directional persistence in cell migration. Nat Protoc 9, 1931–1943 (2014). https://doi.org/10.1038/nprot.2014.131

Metzler R. and Klafter J. The restaurant at the end of the random walk: recent developments in the description of anomalous transport by fractional dynamics. Phys. A: Math. Gen. (2014) 37 R161. http://dx.doi.org/10.1088/0305-4470/37/31/R01

We have clarified this better in the material and method section (lines 5-8, page 19) and added another reference.

2. Figure 4E: the centrosome reporter seems to give a quite high background (derived from other cells?). At the very first picture many cyan dots can be seen. How did the authors identify the centrosome in microglia?

The driver (csf1ra:GAL4-VP16) used in this experiment allows expression of the centrosomal marker not only in microglia and macrophages but also in other cell types such as epithelial cells and pigment cells. As tracking is done in 3D we are able to follow only the microglial centrosome. We have clarified this in the text by adding the following sentence:

“We then tracked the microglial centrosome in 3D and found that in these cells, this organelle is highly dynamic, moving..” (page 5, lines 34 and 35)

3. Figure 5: quantification of EB3/CETN4 migration events are missing.

In Figure 5 we show that the centrosome and the EB3 MTOC colocalize. Thus, quantification of centrosomal movements -in figure 4- provides also information about the EB3-MTOC.

4. Figure 5B: higher magnification picture would help to see MTs growing in the direction of phagosomes.

We thank the reviewer for this suggestion, but the fact that MTs grow in the direction of phagosomes is already shown in the video (Figure 5 – Video 2) we provide. This video shows directional MT polymerization towards forming phagosomes.

5. Figure 7: the authors claim that microglia with two centrosomes are more efficient phagocytes. They quantify the engulfment rate, which is significantly higher in the presence of two centrosomes. This looks very convincing. In the text they state that 'phagocytic events occurring simultaneously at the tip of two different branches, with each branch having a centrosome in the vicinity of the newly formed phagosome'. It would be nice to quantify the number of events, which show two phagocytic events simultaneously vs. consecutive phagocytic events.

We agree with the reviewer; however, simultaneous engulfment events are usually not seen in wt, and in double-centrosomal microglia, they are rare as they depend on the localization of dying neurons relative to the microglial cell and on engulfment happening at the same time by chance. To clarify this point, we have modified the text in the following way:

“Live imaging of double-centrosome microglia revealed that occasionally phagocytic events can also occur simultaneously at the tip of two different branches, with each branch having a centrosome in the vicinity of the newly formed phagosome (Figure 7B)” (page 7 lines 29-31).”

Reviewer #3 (Recommendations for the authors):Figure 1: Can the authors comment on what types of neurons are being engulfed? PVN, interneurons, etc?

According to Robles et al., 2011, most neurons in the optic tectum of the zebrafish become glutamatergic and GABAergic while a minority becomes cholinergic. We are happy to include this information and citation, if necessary.

Figure 4: A further discussion of the so-called 'aborted' phagocytic attempts would be useful. Is this a type of surveillance? Do they abort because the centrosome did not move, or does the centrosome only move when it senses an 'I'm really dead' signal from the neuron?

We thank the reviewer for this comment and are happy to further clarify this point. Our data show that during aborted attempts the centrosome is randomly localized, suggesting that there might not be sufficient signaling to attract the centrosome towards these unstable phagocytic cups. In line with this, when we perturb signaling, the centrosome remains within the cell soma and fails to migrate into branches. We added these observations to the revised manuscript:

“Interestingly, we have found that the centrosome does not migrate towards aborted phagocytic attempts and we speculate that this could be due to low DAG-signaling. In line with this, drug experiments show that when signaling is perturbed, the centrosome remains mostly within the cell soma and phagocytosis is reduced.” (page 9, lines 18-21)

Figure 8: Representative images are lacking for PMA and PLCi treatment. What happens to microtubules and centrosome localization in these phenotypes? Figure 8D is only the mpeg reporter.

We thank the reviewer for this suggestion. We have now added the DMSO and PLCi treatments to Video 8, that previously showed the microglia centrosome in PMA treatment alone. This video now shows the microglia centrosome all three treatments. Moreover, analyses in Figure 8B, C, G and H all refer to centrosome behavior in microglia during these treatments.

Discussion: I'm not a huge fan of calling many different kinds of things a synapse. 'immunologic synapse is by now a well used term, but a 'phagocytic synapse' feels like a step too far, especially for the neuroscience community.

We agree with this reviewer, however, this term has been already used in several reviews to indicate this cell-cell interphase and is now adopted by many. Here are some citations that we have added to the revised manuscript:

Niedergang F, Di Bartolo V, Alcover A. Comparative Anatomy of Phagocytic and Immunological Synapses. Front Immunol. 2016 Jan 28;7:18.

Barth ND, Marwick JA, Vendrell M, Rossi AG, Dransfield I. The "Phagocytic Synapse" and Clearance of Apoptotic Cells. Front Immunol. 2017 Dec 4;8:1708.

Kelley SM, Ravichandran KS. Putting the brakes on phagocytosis: "don't-eat-me" signaling in physiology and disease. EMBO Rep. 2021 Jun 4;22(6):e52564.